# PF3plat: Pose-Free Feed-Forward 3D Gaussian Splatting for Novel View Synthesis

**Sunghwan Hong** [* 1]  **Jaewoo Jung** [* 1]  **Heeseong Shin** [1]  **Jisang Han** [1]  **Jiaolong Yang** [† 2]  **Chong Luo** [† 2]
**Seungryong Kim** [† 1]

## Abstract

We tackle the problem of view synthesis from sparse, unposed images in a single feed-forward pass. Our method builds on 3DGS and relaxes common requirements such as dense views, accurate camera poses or depth, and large image overlaps. However, the main challenge arises from the parametrization of pixel-aligned 3D Gaussians, as their misalignments inevitably yield noisy or sparse gradients that destabilize training. To address this, we leverage pretrained monocular depth estimation and visual correspondence networks for coarse alignment, then refine depth and pose via lightweight learnable modules. We further estimate geometry confidence scores, driven by aggregated monocular and multi-view depth, to assess the reliability of 3D Gaussian centers and condition the prediction of Gaussian parameters accordingly. Extensive experiments on large-scale real-world datasets confirm that PF3plat achieves state-of-the-art performance across all benchmarks, with ablation studies validating our design choices.

## 1. Introduction

In recent years, 3D reconstruction and view synthesis have garnered significant attention, particularly with the emergence of NeRF (Mildenhall et al., 2021) and 3DGS (Kerbl et al., 2023). These advancements have enabled high-quality 3D reconstruction and view synthesis. However, many existing methods rely on stringent assumptions, such as dense image views (Yu et al., 2024; Barron et al., 2021; 2022), accurate camera poses (Charatan et al., 2023; Chen et al.,

2024), and substantial image overlaps (Yu et al., 2021; Johari et al., 2022), which limit their practical applicability. In real-world scenarios, casually captured images contain sparse and distant viewpoints, and lack precise camera poses, making it impractical to assume densely captured views with accurate camera poses.

To address some of these limitations, recent efforts (Yu et al., 2021; Johari et al., 2022; Chen et al., 2021; Yang et al., 2023) have introduced generalized view synthesis frameworks capable of performing single feed-forward novel view synthesis from sparse images with minimal overlaps (Du et al., 2023; Xu et al., 2023a). Among these methods, particularly those utilizing 3DGS (Charatan et al., 2023; Chen et al., 2024), have demonstrated remarkable rendering speed and efficiency, alongside impressive reconstruction and view synthesis quality, highlighting the potential of 3D Gaussian-based representations. However, they still depend on accurate camera poses, which are challenging to acquire in sparse settings, thereby restricting their practical use.

More recently, pose-free generalized view synthesis frameworks (Chen & Lee, 2023; Fan et al., 2023; Jiang et al., 2023; Smith et al., 2023; Hong et al., 2024) have been introduced to decouple view synthesis from camera poses. Given a set of unposed images, these frameworks aim to jointly learn radiance fields and 3D geometry without relying on additional data, such as ground-truth camera pose. The learned radiance fields and geometry can then be inferred through trained neural networks, enabling single feed-forward inference. While these pioneering efforts enhance practicality, their performance remains unsatisfactory and their slow rendering speeds (Chen & Lee, 2023; Fan et al., 2023; Jiang et al., 2023; Smith et al., 2023; Hong et al., 2024) remain unresolved.

In this work, we propose **PF3plat** (Pose-Free Feed-Forward 3D Gaussian Splatting), a novel framework for fast and photorealistic *view synthesis* from *unposed images* in a single *feed-forward pass*. Our approach leverages the efficiency and high-quality reconstruction capabilities of pixel-aligned 3DGS (Charatan et al., 2023; Szymanowicz et al., 2024), while relaxing common assumptions such as dense image views, accurate camera poses, scene-specific optimization

---

*Equal contribution  [1]KAIST AI [2]Microsoft Research Asia. Correspondence to: Jiaolong Yang <jiaoyan@microsoft.com>, Chong Luo <cluo@microsoft.com>, Seungryong Kim <seungryong.kim@kaist.ac.kr>.

*Proceedings of the $42^{nd}$ International Conference on Machine Learning*, Vancouver, Canada. PMLR 267, 2025. Copyright 2025 by the author(s).

and substantial image overlaps. However, a unique challenge emerges in the parametrization of pixel-aligned 3DGS. In scenarios lacking the aforementioned assumptions, inaccuracies in localizing 3D Gaussian centers can lead to noisy or sparse gradients, destabilizing training and hindering convergence. Moreover, unlike scene-specific optimization approaches (Fu et al., 2023; Fan et al., 2024), which can iteratively rectify errors at inference time, typical learning-based feed-forward frameworks are unable to benefit from such iterative refinements.

To mitigate these issues, we leverage pre-trained monocular depth estimation (Piccinelli et al., 2024) and visual correspondence (Lindenberger et al., 2023) models to achieve a coarse alignment of 3D Gaussians to promote a stable learning process. Subsequently, we introduce learnable modules designed to refine the depth and pose estimates from the coarse alignment to enhance the quality of 3D reconstruction and view synthesis. These modules are geometry-aware and lightweight, since we leverage features from the depth network and avoid direct fine-tuning. These refined depth and pose estimates are then used to implement geometry-aware confidence scores to assess the reliability of 3D Gaussian centers, conditioning the prediction of Gaussian parameters such as opacity, covariance, and color.

Our extensive evaluations on large-scale real-world indoor and outdoor datasets (Liu et al., 2021; Zhou et al., 2018; Ling et al., 2024) demonstrate that PF3plat sets a new state-of-the-art across all benchmarks. Comprehensive ablation studies validate our design choices, confirming that our framework provides a fast and high-performance solution for pose-free generalizable novel view synthesis. We summarize our contributions below:

- We propose PF3plat, a feed-forward network that reconstructs 3D scenes, parameterized by 3D Gaussians, from sparse, unposed views without requiring ground-truth depth or pose at either training or inference phase.

- We propose a two-stage pipeline, coarse and fine alignment, to address the unique challenges of pixel-aligned 3D Gaussian parameterization and further enhance the reconstruction and view synthesis quality, respectively.

- Using large-scale real-world indoor and outdoor datsets, we show that our method outperforms existing works in terms of rendered image quality and the camera pose accuracy.

## 2. Related Work

**Generalizable View Synthesis from Unposed Imagery.** Several innovative efforts have addressed the joint learning of camera pose and radiance fields within NeRF-based frameworks. Starting with BARF (Lin et al., 2021), sub-

sequent research (Jeong et al., 2021; Wang et al., 2021; Bian et al., 2023; Truong et al., 2023b) has expanded upon this foundation. Notably, the use of 3D Gaussians as dynamic scene representations has led to significant advancements: Fu et al. (2023) progressively enlarges 3D Gaussians by learning transformations between consecutive frames, SplaTAM(Keetha et al., 2024) utilizes RGB-D sequences and silhouette masks to jointly update Gaussian parameters and camera poses, and InstantSplat (Fan et al., 2024) optimizes 3D Gaussians rapidly for scene reconstruction and view synthesis. Among these, methods like DBARF (Chen & Lee, 2023), FlowCAM (Smith et al., 2023), CoPoN-eRF (Hong et al., 2023), and GGRt (Li et al., 2024) aim to determine camera pose and radiance fields in a single feed-forward pass. Concurrently, Splatt3R (Smart et al., 2024) and NoPoSplat (Ye et al., 2024) leverage pre-trained 3D reconstruction models (Leroy et al., 2024) to relax certain assumptions. However, these methods either target two-view scenarios or rely on additional supervision (e.g., ground-truth depth and pose) during training.

**Monocular Depth and Correspondence Estimation** Monocular depth estimation and visual correspondence estimation are fundamental computer vision tasks that have been extensively researched over several decades. Recent advancements (Yin et al., 2023; Piccinelli et al., 2024; Ke et al., 2024; Yang et al., 2024) in monocular depth estimation have greatly matured these models, thereby expanding the capabilities of numerous 3D vision applications. Likewise, visual correspondence estimation has undergone remarkable progress since the emergence of deep neural networks. Modern approaches have optimized each step, ranging from 2D descriptor extraction (Yi et al., 2016; DeTone et al., 2018) and 3D descriptor extraction (Yew & Lee, 2018; Choy et al., 2019), to sparse and dense matching (Hong & Kim, 2021; Cho et al., 2021; 2022; Hong et al., 2022a;b; 2023; Sun et al., 2021; Edstedt et al., 2024) and outlier filtering (Barath et al., 2019; Wei et al., 2023), surpassing traditional methods in many scenarios.

**Learning-based 3D Reconstruction.** Meanwhile, learning-based 3D reconstruction methods (Wang et al., 2023; Leroy et al., 2024; Wang et al., 2024) have garnered significant attention for their performance in sparse-view reconstruction, forming a robust foundation for tasks such as camera pose estimation and tracking. In contrast, our approach simultaneously tackles 3D reconstruction and view synthesis without requiring lengthy radiance field optimization at inference, enabling more direct and efficient inference.

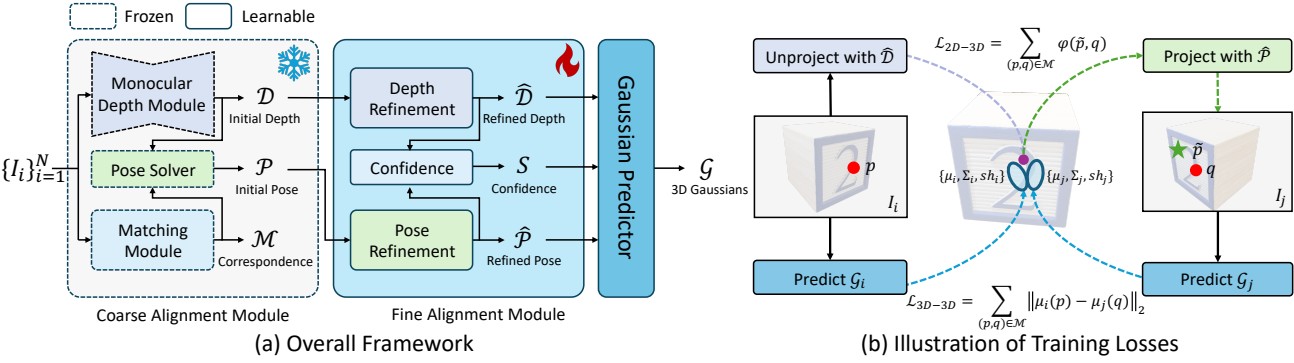

(a) Overall Framework

(b) Illustration of Training Losses

*Figure 1.* **Overall architecture and loss of the proposed method.** (a) Given a set of unposed images and their camera intrinsics, our method aligns the 3D Gaussians using a coarse-to-fine strategy. (b) In addition to photometric loss, we enforce 3D Gaussian consistency by ensuring they are placed on the same object surface through 2D-3D and 3D-3D consistency losses.

## 3. Method

### 3.1. Problem Formulation

Our objective is to reconstruct a 3D scene from a set of $N$ unposed images $\{I_i\}_{i=1}^{N}$ with $I_i \in \mathbb{R}^{H \times W \times 3}$ and corresponding camera intrinsic $K_i$, to synthesize photo-realistic images $\hat{I}_t$ from novel viewpoints in a single feed-forward pass. Note that, in line with existing pose-free view synthesis methods (Fu et al., 2023; Ye et al., 2024; Hong et al., 2024; Chen & Lee, 2023; Smith et al., 2023), we assume camera intrinsic parameters are available, as they are generally available from modern devices (Arnold et al., 2022). To render, we output the depth maps $\mathcal{D}_i \in \mathbb{R}^{H \times W}$ for each image $I_i$, along with their corresponding camera poses $\mathcal{P}_i \in \mathbb{R}^{3 \times 4}$, consisting of a rotation matrix $R_i \in \mathbb{R}^{3 \times 3}$ and a translation vector $t_i \in \mathbb{R}^{3 \times 1}$. Additionally, we compute a set of pixel-aligned 3D Gaussians denoted as $\mathcal{G} = \{\mu_i, \sigma_i, \Sigma_i, c_i\}_{i=1}^{N}$. Here, $\mu_i(p) \in \mathbb{R}^3$ indicates the 3D Gaussian center derived from the depth $\mathcal{D}_i(p)$, camera pose $\mathcal{P}_i$, and camera intrinsic $K_i$, where $p \in \mathbb{R}^{H \times W}$ represents each pixel. Finally, the opacity is represented by $\sigma_i(p) \in [0, 1)$, $\Sigma_i(p) \in \mathbb{R}^{3 \times 3}$ is the covariance matrix, and the color is encoded using spherical harmonics $c_i(p) \in \mathbb{R}^{3(L+1)}$, where $L$ is the order of the spherical harmonics.

### 3.2. PF3plat: Pose-Free Feed-Forward 3D Gaussian Splatting

#### 3.2.1. COARSE ALIGNMENT OF 3D GAUSSIANS

Despite numerous advantages such as speed, efficiency, and high-quality reconstruction and view synthesis (Charatan et al., 2023; Chen et al., 2024), pixel-aligned 3D Gaussians poses certain challenges. Unlike previous methods for generalized novel view synthesis that utilize implicit representations (Chen & Lee, 2023; Smith et al., 2023; Hong et al., 2024) and benefit from the interpolation capabilities of neural networks, our approach is challenged by the explicit

nature of this representation. Specifically, inaccuracies in localizations of 3D Gaussian centers makes it highly vulnerable to noisy and sparse graidents, which cannot be easily compensated.

Such misalignments and sprase gradients can either cause severe performance degradation or disrupt the learning process. This issue is particularly exacerbated when wide-baseline images are given as input or the absence of ground-truth pose or depth prevents alignments of 3D Gaussians. Without effectively addressing these challenges, we find the problem becomes nearly intractable, as we demonstrate in Tab. 4. A possible solution to mitigate this issue is to empoloy iterative scene-specific optimization steps (Fu et al., 2023) or to assume ground-truth camera poses or depth as a guidance for stable learning process (Ye et al., 2024). However, these solutions are incompatible with our goal of achieving a single feed-forward process with training solely from unposed images. Therefore, overcoming these limitations requires a novel strategy that can handle depth and pose ambiguities while maintaining efficiency in a feed-forward manner.

To this end, we propose to provide coarse alignment of 3D Gaussians. We employ off-the-shelf models (Piccinelli et al., 2024; Lindenberger et al., 2023) to estimate initial depths $\mathcal{D}_i$ and camera poses $\mathcal{P}_i$ for our images $I_i$, while other variants can also be leveraged, as we show in supplementary material. Specifically, given depth maps $\mathcal{D}_i$ and sets of correspondences $\mathcal{M}_{ij}$ and their confidence values $\mathcal{C}_{ij}$ acquired from each pairwise combinations of images, *e.g.*, $(I_i, I_j)$, where $i, j$ refer to image indices, we use a robust solver (Fischler & Bolles, 1981; Li et al., 2012) to estimate the relative poses $\mathcal{P}_{ij}$ between image pairs. Integrating these components, we provide the necessary coarse alignment to promote stabilizing the learning process and serve as a strong foundation for further enhancements.

### 3.2.2. MULTI-VIEW CONSISTENT DEPTH ESTIMATION

While pre-trained monocular depth models (Wang et al., 2023; Leroy et al., 2024; Piccinelli et al., 2024; Yin et al., 2023) can offer powerful 3D geometry priors, inherent limitations of these models, namely, inconsistent scales among predictions, still remain unaddressed. This requires further adjustments to ensure multiview consistency across predictions. To overcome this challenge, we aim to refine the predicted depths and camera poses obtained from coarse alignment in a fully learnable and differentiable manner.

Our refinement module includes a pixel-wise depth offset estimation that uses the feature maps $F_i$ from the depth network (Piccinelli et al., 2024) as the sole input and processes them through a series of self-attention operations, making it lightweight and geometry-aware (Xu et al., 2023b). The process is defined as:

$$\Delta\delta_i = \phi_{\text{mlp}}(\mathcal{T}_{\text{depth}}(F_i)), \qquad \hat{\mathcal{D}}_i = \mathcal{D}_i + \Delta\delta_i, \quad (1)$$

where $\phi_{\text{mlp}}(\cdot)$ is a linear projection, $\mathcal{T}_{\text{depth}}$ is a deep Transformer architecture and $\Delta\delta_i$ is the pixel-wise depth offset. This extension promotes consistency across views and enhances performance without relying on explicit cross-attention. Instead, it leverages supervision signals derived from pixel-aligned 3D Gaussians that connect the information across views, and leverage them for the novel view synthesis task (Zhou et al., 2017) and our loss functions, which are detailed in Section 3.3. Additionally, we avoid fine-tuning the entire depth network, thereby reducing computational costs and mitigating the risk of catastrophic forgetting.

### 3.2.3. CAMERA POSE REFINEMENT

Here, we further refine camera poses to enhance reconstruction and view synthesis quality. Initially, we replace the estimated relative poses $\mathcal{P}_{ij}$ with newly computed camera poses $\hat{\mathcal{P}}_{ij}$ derived from following the similar process in coarse alignment and using the previously obtained refined depths $\hat{\mathcal{D}}_i$. We then introduce a learnable camera pose refinement module that estimates rotation and translation offsets. To streamline this process, we first utilize a fully differentiable transformation synchronization operation that takes $\hat{\mathcal{P}}_{ij}$ and $\mathcal{C}_{ij}$ as inputs. Using power iterations (El Banani et al., 2023), this operation efficiently recovers the absolute poses $\hat{\mathcal{P}}_i$ prior to the refinement module.

Next, we convert the absolute poses $\hat{\mathcal{P}}_i$ into Plücker coordinates (Sitzmann et al., 2021), defined as $r = (\text{d}, \text{o} \times \text{d}) \in \mathbb{R}^6$, where d represents the camera direction and o denotes the camera origin. These coordinates, along with the feature maps $F_i \in \mathbb{R}^{h \times w \times d}$ and a pose token $\mathcal{P}_{\text{CLS}} \in \mathbb{R}^d$, are input into a series of self- and cross-attention layers. In our approach, we designate $\hat{\mathcal{P}}_1$ as the reference world space and update only the other pose estimates. The resulting pose to-

ken is then transformed into 6D rotations (Zhou et al., 2019) and translations, which are added to the initial camera poses to estimate the rotation and translation offsets. Formally, these are defined as:

$$\hat{\mathcal{P}}_{\text{CLS}} = \mathcal{T}_{\text{pose}}([F_i, \mathcal{P}_{\text{CLS}}, r] + E_{\text{pos}}),$$
$$\Delta R_i, \Delta t_i = \phi_{\text{rot}}(\hat{\mathcal{P}}_{\text{CLS}}), \phi_{\text{trans}}(\hat{\mathcal{P}}_{\text{CLS}}), \quad (2)$$
$$\hat{R}_i, \hat{t}_i \Leftarrow \hat{R}_i + \Delta R_i, \hat{t} + \Delta t,$$

where $\Delta R, \Delta t$ are the pose offsets, and $E_{\text{pos}}$ is positional embedding.

### 3.2.4. 3D GAUSSIAN PARAMTER PREDICTIONS

**Multi-View and Guidance Cost Volume Construction and Aggregation.** Using the refined pose and the monocular depth estimates, $\hat{\mathcal{P}}_i$ and $\hat{\mathcal{D}}_i$, we assess the quality of the predictions to obtain confidence scores, to assist predicting 3D Gaussian parameters. To achieve this, we construct both a conventional multi-view stereo cost volume and a guidance cost volume derived from $\mathcal{D}_i$.

Specifically, given the current pose estimates $\hat{\mathcal{P}}_i$, we build a multi-view stereo cost volume $\mathcal{C}_i^{\text{multi}} \in \mathbb{R}^{h \times w \times K}$ following the plane-sweeping approach (Yao et al., 2018; Chen et al., 2021; 2024; An et al., 2024). For each of the $K$ depth candidates within specified near and far ranges along the epipolar lines, we compute matching scores using cosine similarity (Cho et al., 2022; 2021; Hong et al., 2024). Subsequently, to guide the depth localization along the epipolar lines, we construct a guidance cost volume $\mathcal{C}_i^{\text{guide}}$ (Li et al., 2023), where each spatial location is represented by a one-hot vector indicating the depth candidate closest to the monocular depth estimate. We finally obtain an aggregated cost volume $\mathcal{C}_i^{\text{agg}}$ by feeding $\mathcal{C}_i^{\text{multi}}$ and $\mathcal{C}_i^{\text{guide}}$ to a series of cross-attention layers to update the multi-view cost volume guided by a guidance cost volume. We define the process as following:

$$\mathcal{C}_i^{\text{agg}} = \mathcal{T}_{agg}(\mathcal{C}_i^{\text{multi}}, \mathcal{C}_i^{\text{guide}}), \quad (3)$$

where $\mathcal{T}(\cdot)$ is a deep transformer architecture that computes cross-attention between inputs.

**Geometry-aware Confidence Estimation.** Using the aggregated cost volume $\mathcal{C}_i^{\text{agg}}$, we apply a softmax function (Xu et al., 2022; Hong et al., 2023) along the $K$ dimension to obtain a matching distribution. We then extract the maximum value from this distribution to derive a confidence score, $S_{\text{geo}}$, which assesses the quality of the estimated camera pose and depth. Formally, this is defined as:

$$S_i^{\text{geo}} = \max_{k \in \{1,2,\ldots,K\}} \text{softmax}(\mathcal{C}_i^{\text{agg}})(k). \quad (4)$$

These confidence scores assess the reliability of the predicted 3D Gaussian centers, where high confidence indicates

accurate localization and low confidence suggests potential inaccuracies due to noise or misalignment. To condition the prediction of Gaussian parameters, such as opacity, covariance, and color, we incorporate $S_{\text{geo}}$ as additional input.

**3D Gaussian Parameters.** Finally, using the inputs $[I_i, \hat{D}_i, F_i, S_i^{\text{geo}}]$, we compute the opacity $\sigma_i$ through small convolutional layers, derive the covariances from the estimated rotations and scales, and obtain the color from the estimated SH coefficients. A key idea of our approach is that $S^{\text{geo}}$ enables supervision signals to flow from the Gaussian parameters back to the depth and pose estimates. This feedback loop enhances the accuracy of both depth and pose estimations, resulting in more consistent and reliable 3D reconstructions.

### 3.3. Loss Function

**Reconstruction Loss.** We combine photometric loss, defined as the L2 loss between the rendered and target images, as well as SSIM (Wang et al., 2004) loss $\mathcal{L}_{\text{SSIM}}$ and LPIPS (Zhang et al., 2018) loss $\mathcal{L}_{\text{LPIPS}}$ to form our reconstruction loss $\mathcal{L}_{\text{img}}$.

**2D-3D Consistency Loss.** We identify that provided good coarse alignments, RGB loss is sufficient, as similarly observed in (Ye et al., 2024), but with larger baselines, the training process starts to destabilize. Moreover, one remaining issue with learning solely from the photometric loss is that the gradients are mainly derived from pixel intensity differences, which suffer in textureless regions. To remedy these, we enforce that corresponding points in the set of images $\{I\}_{i=1}^N$ lie on the same object surface, drawing from principles of multi-view geometry (Hartley & Zisserman, 2003).

Formally, using the estimated depths $\hat{D}$, the camera poses $\hat{\mathcal{P}}$, and the correspondence sets $\mathcal{M}$, we can define a geometric consistency loss that penalizes deviations from the multi-view geometric constraints. For each correspondence $(p, q) \in \mathcal{M}_{ij}$ between images $I_i$ and $I_j$, we compute the 3D point from the pixel $p$ and its estimated depth $\hat{D}_i(p)$ using the camera intrinsics. We then transform this to the coordinate frame of $I_j$ using the relative pose $\hat{\mathcal{P}}_{ij}$ and project it back onto the image plane to obtain the predicted correspondence $\tilde{p}$. This is defined as $\mathcal{L}_{\text{2D}-\text{3D}} = \sum_{(p,q)\in\mathcal{M}} \varphi(\tilde{p}, q)$, where $\varphi(\cdot)$ denotes huber loss.

**3D-3D Consistency Loss.** While the multi-view consistent surface loss directly connects each pair of corresponding Gaussians and their centers, guiding the model towards the object's surface, we find that relying solely on this loss can lead to suboptimal convergence, especially in regions with sparse correspondences. To further stabilize and enhance the learning process, we introduce an additional regularization term that minimizes the discrepancies among the centers of the corresponding Gaussians.

Intuitively, this differs from $\mathcal{L}_{\text{2D}-\text{3D}}$ in that, unlike the previous function, which considers the alignment of Gaussian centers from only one side when dealing with pairwise correspondences, the regularization term symmetrically enforces consistency from both sides. Specifically, while the multi-view consistent surface loss projects the Gaussian center from one view to another using the estimated depth and camera pose, *e.g.,* from source to target, the regularization term jointly minimizes the distances between all corresponding Gaussian centers across multiple views. By considering both directions in pairwise correspondences, this approach promotes a more coherent and robust estimation of the object's surface, reducing the influence of outliers and improving convergence during training. This additional regularization can be formally defined as:
$\mathcal{L}_{\text{3D}-\text{3D}} = \sum_{(p,q)\in\mathcal{M}} ||\mu_i(p) - \mu_j(q)||_2$.

**Final Objective Function.** Combining the three loss functions, we define our final objective function: $\mathcal{L}_{\text{img}} + \mathcal{L}_{\text{2D}-\text{3D}} + \lambda_{\text{3D}-\text{3D}}\mathcal{L}_{\text{3D}-\text{3D}}$, where we set $\lambda_{\text{3D}-\text{3D}} = 0.05$.

## 4. Experiments

### 4.1. Implementation Details

We compute attentions using Flash Attention (Dao et al., 2022), and for the Gaussian rasterizer, we follow the method described in (Charatan et al., 2023). Our model is trained on 4 NVIDIA A100 GPU for 50,000 iterations using the Adam optimizer (Kingma, 2014), with a learning rate set to $8 \times 10^{-4}$ and a batch size of 9 per each GPU, which takes approximately two days. For training on the RealEstate10K and ACID datasets, we gradually increase the frame distance between $I_1$ and $I_2$ as training progresses, initially setting the frame distance to 15 and gradually increasing it to 75. For the DL3DV dataset, we start with a frame distance of 5 and increase it to 10. The target view is randomly sampled within this range. The code and pretrained weights will be made publicly available.

### 4.2. Experimental Setting

**Datasets.** We train and evaluate our method on three large-scale datasets: RealEstate10K (Zhou et al., 2018), a collection of both indoor and outdoor scenes; ACID (Liu et al., 2021), a dataset focusing on outdoor coastal scenes; and DL3DV (Ling et al., 2024), which includes diverse real-world indoor and outdoor environments. For RealEstate10K, due to some unavailable videos on YouTube, we use a subset of the full dataset, comprising a training set of 21,618 scenes and a test set of 7,200 scenes. For ACID, we train on 10,935 scenes and evaluate on 1,893 scenes. Lastly, for DL3DV, we train on 10,510 different scenes and evaluate on the standard benchmark set of 140 scenes for testing (Ling et al., 2024).

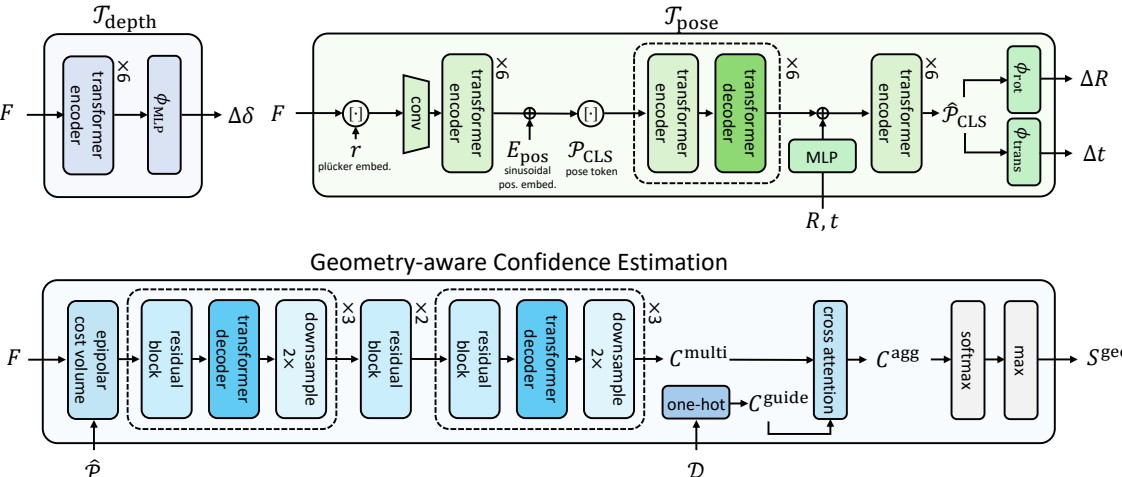

*Figure 2.* **Proposed refinement and confidence estimation modules.** In our Fine Alignment module, we refine depth and pose to improve 3D reconstruction and view synthesis quality, alongside estimating confidence to assess the reliability of predicted 3D Gaussian centers.

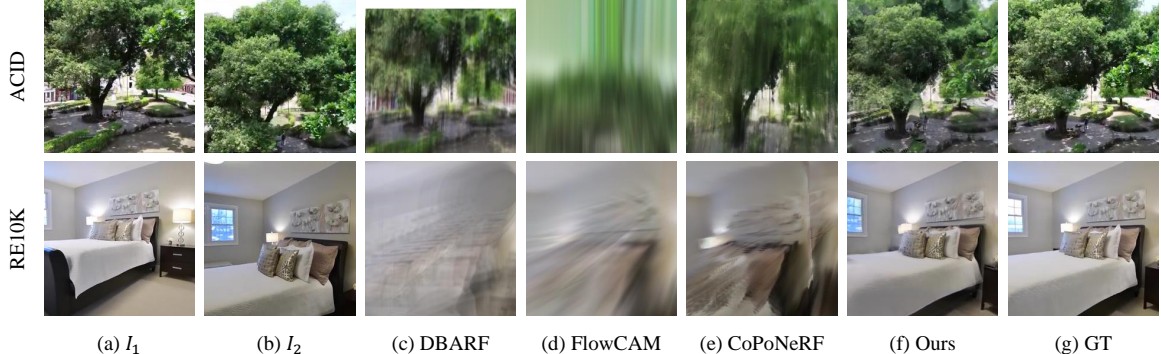

| (a) $I_1$ | (b) $I_2$ | (c) DBARF | (d) FlowCAM | (e) CoPoNeRF | (f) Ours | (g) GT |

*Figure 3.* **Qualitative Comparison on RealEstate-10K and ACID.** Given two context views (a) and (b), we compare novel view rendering results.

**Baselines.** Following (Hong et al., 2024), we evaluate our method on two tasks: novel-view synthesis and camera pose estimation. For novel view synthesis, we compare our approach against established generalized NeRF and 3DGS variants, including PixelNeRF (Yu et al., 2021), (Du et al., 2023), PixelSplat (Charatan et al., 2023), and MVSPlat (Chen et al., 2024). It is important to note that these methods assume GT camera poses during training and inference, so we include them only for reference. Our primary comparisons focus on existing pose-free generalized novel view synthesis methods, such as DBARF (Chen & Lee, 2023), FlowCAM (Smith et al., 2023), and CoPoNeRF (Hong et al., 2024).

For camera pose estimation, we evaluate against correspondence-based pose estimation methods (e.g., SfM), including SP+SG (DeTone et al., 2018; Sarlin et al., 2020), PDC-Net+ (Truong et al., 2023a), DUSt3R (Wang et al., 2023), and MASt3R (Leroy et al., 2024). Due to absence of GT depth in our datasets, we leverage their pre-trained weights and direct comparison is avoided. Additionally,

we compare with direct pose regression methods such as 8ViT (Rockwell et al., 2022), RelPose (Zhang et al., 2022). Our main comparisons include pose-free view synthesis approaches, including DBARF (Chen & Lee, 2023), FlowCAM (Smith et al., 2023), and CoPoNeRF (Hong et al., 2024).

**Evaluation Protocol** For evaluation, we follow the protocol outlined by (Hong et al., 2024) using unposed triplet images $(I_1, I_2, I_t)$, with the test set divided into small, middle, and large based on the extent of overlap between $I_1$ and $I_2$. We also provide multi-view evaluation in Tab. 5c. We provide further details in the supplementary materials.

### 4.3. Experimental Results

**RealEstate-10K & ACID.** Tab.1 summarizes the performance for the novel view synthesis task, while Tab.2 reports the results for pose estimation. From the results in Tab. 1, our method significantly outperforms previous pose-free generalizable methods (Chen & Lee, 2023; Smith et al.,

*Table 1.* **Novel View Synthesis Performance on RealEstate-10K and ACID.** Gray entries indicate methods that use ground truth camera poses during evaluation and are not directly comparable.

**RealEstate-10K**

| Pose-Free | Method | Avg | | | | Small | | | | Medium | | | | Large | | | |
|---|---|---|---|---|---|---|---|---|---|---|---|---|---|---|---|---|---|
| | | PSNR | LPIPS | SSIM | MSE | PSNR | LPIPS | SSIM | MSE | PSNR | LPIPS | SSIM | MSE | PSNR | LPIPS | SSIM | MSE |
| ✗ | PixelNeRF | 14.438 | 0.577 | 0.467 | 0.047 | 13.126 | 0.639 | 0.466 | 0.058 | 13.999 | 0.582 | 0.462 | 0.042 | 15.448 | 0.479 | 0.470 | 0.031 |
| | Du *et al.* | 21.833 | 0.294 | 0.736 | 0.011 | 18.733 | 0.378 | 0.661 | 0.018 | 22.552 | 0.263 | 0.764 | 0.008 | 26.199 | 0.182 | 0.836 | 0.004 |
| | PixelSplat | 24.788 | 0.176 | 0.820 | 0.009 | 21.222 | 0.246 | 0.742 | 0.013 | 26.106 | 0.862 | 0.135 | 0.004 | 29.545 | 0.916 | 0.092 | 0.003 |
| | MVSplat | 25.054 | 0.157 | 0.827 | 0.008 | 21.029 | 0.226 | 0.747 | 0.013 | 26.369 | 0.116 | 0.874 | 0.004 | 30.516 | 0.074 | 0.926 | 0.002 |
| ✓ | DBARF | 14.789 | 0.490 | 0.570 | 0.033 | 13.453 | 0.563 | 0.522 | 0.045 | 15.201 | 0.487 | 0.560 | 0.030 | 16.615 | 0.380 | 0.648 | 0.022 |
| | FlowCAM | 18.242 | 0.597 | 0.455 | 0.023 | 15.435 | 0.528 | 0.570 | 0.034 | 18.481 | 0.592 | 0.441 | 0.18 | 22.418 | 0.707 | 0.287 | 0.009 |
| | CoPoNeRF | 19.536 | 0.398 | 0.638 | 0.016 | 17.153 | 0.459 | 0.577 | 0.025 | 19.965 | 0.343 | 0.645 | 0.013 | 22.542 | 0.250 | 0.724 | 0.008 |
| | Ours | 23.589 | 0.181 | 0.782 | 0.010 | 19.998 | 0.244 | 0.700 | 0.015 | 24.073 | 0.155 | 0.819 | 0.006 | 28.834 | 0.098 | 0.889 | 0.003 |

**ACID**

| Pose-Free | Method | Avg | | | | Small | | | | Medium | | | | Large | | | |
|---|---|---|---|---|---|---|---|---|---|---|---|---|---|---|---|---|---|
| | | PSNR | LPIPS | SSIM | MSE | PSNR | LPIPS | SSIM | MSE | PSNR | LPIPS | SSIM | MSE | PSNR | LPIPS | SSIM | MSE |
| ✗ | PixelNeRF | 17.160 | 0.527 | 0.496 | 0.029 | 16.996 | 0.528 | 0.487 | 0.030 | 17.228 | 0.534 | 0.501 | 0.029 | 17.229 | 0.522 | 0.500 | 0.028 |
| | Du *et al.* | 25.482 | 0.304 | 0.769 | 0.005 | 25.553 | 0.301 | 0.773 | 0.005 | 25.694 | 0.303 | 0.769 | 0.005 | 25.338 | 0.307 | 0.763 | 0.005 |
| | PixelSplat | 28.336 | 0.157 | 0.834 | 0.004 | 28.142 | 0.164 | 0.827 | 0.004 | 28.650 | 0.148 | 0.846 | 0.003 | 28.306 | 0.157 | 0.833 | 0.004 |
| | MVSplat | 28.252 | 0.157 | 0.829 | 0.004 | 28.085 | 0.164 | 0.820 | 0.004 | 28.571 | 0.148 | 0.843 | 0.003 | 28.203 | 0.156 | 0.828 | 0.004 |
| ✓ | DBARF | 14.189 | 0.452 | 0.537 | 0.038 | 14.306 | 0.503 | 0.541 | 0.037 | 14.253 | 0.457 | 0.538 | 0.038 | 14.086 | 0.419 | 0.534 | 0.039 |
| | FlowCAM | 20.116 | 0.477 | 0.585 | 0.016 | 20.153 | 0.475 | 0.594 | 0.016 | 20.158 | 0.476 | 0.585 | 0.015 | 20.073 | 0.478 | 0.580 | 0.016 |
| | CoPoNeRF | 22.440 | 0.323 | 0.649 | 0.010 | 22.322 | 0.358 | 0.649 | 0.010 | 22.407 | 0.352 | 0.648 | 0.009 | 22.529 | 0.351 | 0.649 | 0.009 |
| | Ours | 25.640 | 0.204 | 0.784 | 0.006 | 25.882 | 0.205 | 0.788 | 0.006 | 25.998 | 0.211 | 0.790 | 0.006 | 25.321 | 0.201 | 0.778 | 0.006 |

*Table 2.* **Pose Estimation Performance on RealEstate-10K and ACID.** Gray entries indicate methods that were not trained on the same dataset due to missing ground-truth data (e.g., depth or correspondences), making them not directly comparable to ours. In other words, we cannot train our approach on their dataset, nor can they train theirs on ours. *: We also include a MASt3R variant that omits iterative pose optimization (roughly taking $\sim$ 10s) and instead relies on a PnP solver to maintain consistency with other baselines.

**RealEstate-10K**

| Task | Additional Info. | Method | Avg | | | | Small | | | | Medium | | | | Large | | | |
|---|---|---|---|---|---|---|---|---|---|---|---|---|---|---|---|---|---|---|
| | | | Rotation | | Translation | | Rotation | | Translation | | Rotation | | Translation | | Rotation | | Translation | |
| | | | Avg(°↓) | Med(°↓) | Avg(°↓) | Med(°↓) | Avg(°↓) | Med(°↓) | Avg(°↓) | Med(°↓) | Avg(°↓) | Med(°↓) | Avg(°↓) | Med(°↓) | Avg(°↓) | Med(°↓) | Avg(°↓) | Med(°↓) |
| SfM | Pose + Depth | SP+SG | 5.605 | 1.301 | 14.89 | 5.058 | 9.793 | 2.270 | 12.55 | 4.638 | 1.789 | 0.969 | 9.295 | 3.279 | 1.416 | 0.847 | 21.42 | 7.190 |
| | | PDC-Net+ | 2.189 | 0.751 | 10.10 | 3.243 | 3.460 | 1.128 | 6.913 | 2.752 | 1.038 | 0.607 | 6.667 | 2.262 | 0.981 | 0.533 | 16.57 | 5.447 |
| | | DUSt3R | 2.527 | 0.814 | 17.45 | 4.131 | 3.856 | 1.157 | 12.23 | 2.899 | 1.650 | 0.733 | 14.00 | 3.650 | 0.957 | 0.476 | 27.30 | 10.27 |
| | | MASt3R | 2.555 | 0.751 | 9.775 | 2.830 | 4.240 | 1.283 | 8.050 | 2.515 | 1.037 | 0.573 | 6.904 | 2.418 | 0.791 | 0.418 | 13.963 | 3.925 |
| | | MASt3R* | 3.392 | 1.455 | 24.346 | 8.997 | 5.048 | 1.954 | 14.232 | 5.472 | 2.045 | 1.261 | 19.574 | 8.581 | 1.576 | 1.059 | 42.385 | 28.390 |
| Pose Estimation | Pose | 8ViT | 12.59 | 6.881 | 90.12 | 88.65 | 12.60 | 6.860 | 91.46 | 91.50 | 12.17 | 6.552 | 82.48 | 82.92 | 12.77 | 7.214 | 91.85 | 88.92 |
| | | RelPose | 8.285 | 3.845 | - | - | 12.10 | 4.803 | - | - | 4.942 | 3.476 | - | - | 4.217 | 2.447 | - | - |
| Pose-Free View Synthesis | ✗ | DBARF | 11.14 | 5.385 | 93.30 | 102.5 | 17.52 | 13.22 | 126.3 | 140.4 | 7.254 | 4.379 | 79.40 | 75.41 | 3.455 | 1.937 | 50.09 | 33.96 |
| | ✗ | FlowCAM | 7.426 | 4.051 | 50.66 | 46.28 | 11.88 | 6.778 | 87.12 | 58.25 | 4.154 | 3.346 | 42.29 | 41.59 | 2.349 | 1.524 | 34.47 | 27.79 |
| | Pose | CoPoNeRF | 3.610 | 1.759 | 12.77 | 7.534 | 5.471 | 2.551 | 11.86 | 5.344 | 2.183 | 1.485 | 10.19 | 5.749 | 1.529 | 0.991 | 15.544 | 7.907 |
| | ✗ | Ours | 1.756 | 0.897 | 9.474 | 4.628 | 2.338 | 1.002 | 7.121 | 4.005 | 1.287 | 1.338 | 9.211 | 4.266 | 1.118 | 0.499 | 13.225 | 5.778 |

**ACID**

| Task | Additional Info. | Method | Avg | | | | Small | | | | Medium | | | | Large | | | |
|---|---|---|---|---|---|---|---|---|---|---|---|---|---|---|---|---|---|---|
| | | | Rotation | | Translation | | Rotation | | Translation | | Rotation | | Translation | | Rotation | | Translation | |
| | | | Avg(°↓) | Med(°↓) | Avg(°↓) | Med(°↓) | Avg(°↓) | Med(°↓) | Avg(°↓) | Med(°↓) | Avg(°↓) | Med(°↓) | Avg(°↓) | Med(°↓) | Avg(°↓) | Med(°↓) | Avg(°↓) | Med(°↓) |
| SfM | Pose + Depth | SP+SG | 4.819 | 1.203 | 20.802 | 6.878 | 10.920 | 2.797 | 22.214 | 7.526 | 3.275 | 1.306 | 16.455 | 5.426 | 1.851 | 0.745 | 22.018 | 7.309 |
| | | PDC-Net+ | 4.830 | 1.742 | 48.409 | 28.258 | 2.520 | 0.579 | 15.664 | 4.215 | 2.378 | 0.688 | 14.940 | 4.301 | 1.953 | 0.636 | 18.447 | 4.357 |
| | | DUSt3R | 5.558 | 1.438 | 50.661 | 36.154 | 6.515 | 1.450 | 51.348 | 39.334 | 4.773 | 1.392 | 49.647 | 35.105 | 5.346 | 1.444 | 50.724 | 35.260 |
| | | MASt3R | 2.320 | 0.625 | 25.325 | 7.334 | 2.223 | 0.647 | 25.382 | 8.107 | 1.977 | 0.613 | 24.460 | 6.635 | 2.544 | 0.613 | 25.697 | 7.099 |
| | | MASt3R* | 3.988 | 1.438 | 45.376 | 25.917 | 4.458 | 1.461 | 45.328 | 27.233 | 3.214 | 1.364 | 45.303 | 27.540 | 4.062 | 1.473 | 45.160 | 24.870 |
| Pose Estimation | Pose | 8ViT | 4.568 | 1.312 | 88.433 | 88.961 | 8.466 | 3.151 | 88.421 | 88.958 | 4.325 | 1.564 | 90.555 | 90.799 | 2.280 | 0.699 | 86.580 | 87.559 |
| | | RelPose | 6.348 | 2.567 | - | - | 10.081 | 4.753 | - | - | 5.801 | 2.803 | - | - | 4.309 | 2.011 | - | - |
| Pose-Free View Synthesis | ✗ | DBARF | 4.681 | 1.421 | 71.711 | 68.892 | 8.721 | 3.205 | 95.149 | 99.490 | 4.424 | 1.685 | 77.324 | 77.291 | 2.303 | 0.859 | 54.523 | 38.829 |
| | ✗ | FlowCAM | 9.001 | 6.749 | 95.405 | 88.133 | 8.663 | 6.675 | 92.130 | 85.846 | 8.778 | 6.589 | 95.444 | 87.308 | 9.305 | 6.898 | 97.392 | 89.359 |
| | Pose | CoPoNeRF | 3.283 | 1.134 | 22.809 | 14.502 | 3.548 | 1.129 | 23.689 | 11.289 | 2.573 | 1.169 | 21.401 | 10.656 | 3.455 | 1.129 | 22.935 | 10.588 |
| | ✗ | Ours | 2.691 | 1.113 | 20.319 | 9.366 | 2.551 | 0.998 | 22.888 | 8.889 | 1.989 | 0.788 | 17.884 | 8.887 | 3.112 | 1.338 | 19.887 | 9.887 |

2023; Hong et al., 2024), achieving a 4.0 dB improvement to CoPoNeRF in PSNR, demonstrating superior reconstruction quality and robustness. We also provide qualitative comparison in Fig. 3. Additionally, our approach also demonstrates superior pose estimation performance on both datasets, even surpassing (Hong et al., 2024) that trains its network with GT camera poses; however, we observe that compared to MASt3R, we slightly fall behind on ACID dataset. This discrepancy may be attributed to the larger scale of scenes, such as coastal landscapes and sky views, or dynamic scenes in ACID which complicates our refinement process and poses challenges for our depth network in estimating the metric depth of the scene.

**DL3DV.** While RealEstate-10K and ACID encompass a variety of indoor and outdoor scenes, RealEstate-10K pre-dominantly includes indoor environments, whereas ACID features numerous dynamic scenes. To comprehensively evaluate our method across a broader spectrum of real-world scenarios, we further assess it on the recently released DL3DV dataset (Ling et al., 2024). The results are summarized in Table 3. From these results, we observe that our method outperforms CoPoNeRF (Hong et al., 2024) by over 5 dB in large-overlap scenarios and by 4 dB in small-overlap scenarios, highlighting the superior accuracy and robustness of our approach in handling diverse and complex environments. This highlights the effectiveness of our method in managing varied scene and object types, reinforcing its applicability for practical view synthesis tasks.

*Table 3.* **Novel View Synthesis and Pose Estimation Performance on DL3DV**. We include PixelSplat and MVSplat for reference only.

| | | DL3DV | | | | | | | | | | | | | |
|---|---|---|---|---|---|---|---|---|---|---|---|---|---|---|---|
| | | Small | | | | | | | Large | | | | | | |
| Pose-Free | Method | PSNR | LPIPS | SSIM | Rotation | | Translation | | PSNR | LPIPS | SSIM | Rotation | | Translation | |
| | | | | | Avg. | Med. | Avg. | Med. | | | | Avg. | Med. | Avg. | Med. |
| ✗ | PixelSplat | 19.427 | 0.342 | 0.582 | - | - | - | - | 22.889 | 0.193 | 0.734 | - | - | - | - |
| ✗ | MVSPlat | 20.849 | 0.230 | 0.680 | - | - | - | - | 24.211 | 0.147 | 0.796 | - | - | - | - |
| ✓ | CoPoNeRF | 15.509 | 0.563 | 0.396 | 13.121 | 6.721 | 44.645 | 30.269 | 17.586 | 0.467 | 0.469 | 5.609 | 2.905 | 17.974 | 12.445 |
| ✓ | Ours | **19.822** | **0.248** | **0.651** | **4.338** | **2.339** | **9.998** | **6.532** | **22.668** | **0.198** | **0.723** | **3.448** | **1.598** | **9.338** | **6.177** |

*Table 4.* **Component ablations on RealEstate10K.**

| | | Avg | | | | | | |
|---|---|---|---|---|---|---|---|---|
| | Components | PSNR | SSIM | LPIPS | Rotation | | Translation | |
| | | | | | Avg. | Med. | Avg. | Med. |
| **(0)** | Baseline | 20.140 | 0.694 | 0.281 | 2.776 | **0.630** | 10.043 | **3.264** |
| **(I)** | PFSplat | **23.589** | **0.782** | **0.181** | **1.756** | 0.897 | **9.474** | 4.628 |
| **(II)** | - Depth Refinement | 22.012 | 0.754 | 0.203 | 2.342 | 1.122 | 9.881 | 4.952 |
| **(III)** | - Pose Refinement | 21.623 | 0.744 | 0.219 | 2.310 | 1.233 | 11.889 | 6.544 |
| **(IV)** | - Geometry Confidence | 21.443 | 0.741 | 0.223 | 2.228 | 1.001 | 11.322 | 5.998 |
| **(V)** | - Corres. Network | N/A | N/A | N/A | N/A | N/A | N/A | N/A |
| **(VI)** | - Mono. Depth Network | 16.132 | 0.511 | 0.405 | 6.990 | 5.329 | 21.328 | 14.432 |
| **(I-I)** | Full F.T. Depth Network | N/A | N/A | N/A | N/A | N/A | N/A | N/A |
| **(I-II)** | Scale/Shift Tuning Depth Network | N/A | N/A | N/A | N/A | N/A | N/A | N/A |
| **(I-III)** | - Tri. Consis. Loss | 19.001 | 0.644 | 0.402 | 5.661 | 2.099 | 18.332 | 10.331 |
| **(I-IV)** | - Regularization Loss | 21.332 | 0.733 | 0.231 | 4.555 | 2.012 | 12.338 | 9.998 |
| **(I-V)** | (I-IV) - Tri. Consis. loss | N/A | N/A | N/A | N/A | N/A | N/A | N/A |

## 4.4. Ablation Study

In this ablation study, we aim to investigate the effectiveness of each component of our method. We first define a baseline model, which combines our depth and pose estimates from coarse alignments with MVSplat (Chen et al., 2024) for 3D Gaussian parameter prediction. We also explore both full fine-tuning and partial fine-tuning strategies for the depth network. Additionally, we report the results of ablation studies on our loss functions. The results are summarized in Tab. 4.

From the results, we find that significant improvement from (**0**). This improvement is further supported by the comparisons from (**I**) to (**IV**) and from (**I-III**) to (**I-IV**), which show performance degradation as each component is removed. While we observe that (**0**) surpasses our final model in terms of the median rotation and translation angular difference, this aligns with a commonly observed trend between learning-based and robust solver-based approaches (Rockwell et al., 2024). Classical methods tend to achieve higher precision, whereas learning-based approaches generally offer greater robustness.

We also demonstrate that without pre-trained weights for the depth and correspondence networks, the training either fails or achieves significantly lower performance. Similar observations are made in (**I-I**), (**I-II**), and (**I-V**), where we identify that directly tuning the depth network or training only with photometric losses leads to failure in the training process. The former issue may arise from overfitting, a common problem when directly manipulating foundation models. With only the photometric loss, we observe that after certain iterations, as the baseline becomes wider, the training loss quickly diverges.

## 4.5. Analysis and More Results

**Comparison to scene-specific optimization methods.** In this study, we also compare our method with InstantSplat (Fan et al., 2024) and CF-3DGS (Fu et al., 2023), as summarized in Tab. 5a. Our approach already surpasses InstantSplat, a method that adopts similar 2-stage approach as ours, but instead of feed-forward inference, it iteratively optimizes the 3D Gaussian parameters. This results highlights the effectiveness of our refinement modules and our design. The performance gap widens further when we adopt a similar test-time optimization (TTO) strategy. By using our predictions as initialization, TTO takes significantly less time than InstantSplat, demonstrating high practicality. Finally, CF-3DGS struggles to find accurate camera poses and suffers in rendering quality, likely because its design does not adequately handle wide-baseline images.

**Inference speed comparisons.** We conduct a comprehensive inference speed comparison between our method and competing approaches using varying numbers of input images, specifically $N \in 2, 6, 12$. For each scenario, we evaluate the time required to render 1, 3, and 5 views. The results, summarized in Tab. 5b, show that our approach is generally faster than existing methods. However, for $N = 12$, our inference speed is slower than that of DBARF, as our method involves estimating camera poses via a robust solver for every pairwise combination. Despite this overhead, our approach gains a significant advantage as the number of rendered views increases, due to the efficient rendering capabilities of 3DGS once the scene has been fully reconstructed. Finally, we provide the inference time of each of our components: overall inference time, UniDepth processing time, and decoder time. Given 2, 6, 12 views and to render a single target view, it takes 0.251, 0.832, 1.535 seconds for UniDepth inference, while rendering takes approximately consistent 0.00247 seconds.

**Extending to N-views.** In practical scenarios, more than two views ($N > 2$) are commonly used. Moreover, unlike (Ye et al., 2024), our method naturally supports extension to $N$-views, without needing to train our network again. Here, we demonstrate that our method can process multiple views and render $I_t$. We input $N$ views into our network to obtain $\hat{\mathcal{P}}_i$, $\hat{\mathcal{D}}_i$, and $(\mu_i, \sigma_i, \Sigma_i, c_i)$. Following a similar approach to (Chen & Lee, 2023), we select the top-$k$ nearby views using $\hat{\mathcal{P}}_i$ and render $\hat{I}_t$ to compare with the ground

*Table 5.* **More Analysis and Results.**

(a) **Performance and speed comparisons on RealEstate-10K against per-scene optimization methods.**

| Method | PSNR | SSIM | LPIPS | Rot. (°↓) | | Trans. (°↓) | | Time (s) |
|---|---|---|---|---|---|---|---|---|
| | | | | Avg. | Med. | Avg. | Med. | |
| InstantSplat | 23.079 | 0.777 | 0.182 | 2.693 | 0.882 | 11.866 | **3.094** | 53 |
| CF-3DGS | 14.024 | 0.455 | 0.450 | 13.278 | 8.486 | 106.397 | 106.337 | 25 |
| Ours | 23.589 | 0.782 | 0.181 | 1.756 | 0.897 | 9.474 | 4.628 | **0.390** |
| Ours + TTO | **24.689** | **0.798** | **0.167** | **1.662** | **0.871** | **8.998** | 4.311 | 24 |

(b) **Speed comparisons between pose-free generalizable view synthesis models.** Times are measured in seconds.

| Method | 2 views | | | 6 views | | | 12 views | | |
|---|---|---|---|---|---|---|---|---|---|
| | 1 view | 3 view | 5 view | 1 view | 3 view | 5 view | 1 view | 3 view | 5 view |
| DBARF | 1.456 | 4.562 | 8.177 | 2.965 | 7.288 | 13.780 | **4.009** | 10.493 | 17.50 |
| FlowCAM | 4.010 | 7.020 | 10.13 | 9.564 | 23.718 | 34.000 | 14.34 | 23.44 | 48.55 |
| CoPoNeRF | 17.29 | 33.78 | 54.52 | N/A | N/A | N/A | N/A | N/A | N/A |
| Ours | **0.390** | **0.392** | **0.394** | **2.054** | **2.056** | **2.058** | 5.725 | **5.727** | **5.729** |

(c) **RealEstate10K 6, 12 input views.**

| Method | 6 views | | | | 12 views | | | |
|---|---|---|---|---|---|---|---|---|
| | PSNR | SSIM | LPIPS | ATE | PSNR | SSIM | LPIPS | ATE |
| DBARF | 23.91662 | 0.7837 | 0.2226 | 0.0101166 | 24.1798 | 0.7906 | 0.2186 | 0.0048777 |
| FlowCAM | 24.6660 | 0.8259 | 0.2332 | 0.0022202 | 25.2290 | 0.8406 | 0.2169 | 0.0012655 |
| Ours | **27.0284** | **0.8788** | **0.1158** | **0.0010048** | **28.1334** | **0.9934** | **0.0988** | **0.0004228** |

(d) **Cross-dataset Evaluation.**

| Method | RealEstate10K → DL3DV | | | | | | | | DL3DV → RealEstate10K | | | | | | | |
|---|---|---|---|---|---|---|---|---|---|---|---|---|---|---|---|---|
| | PSNR | SSIM | LPIPS | Rot.(°↓) | | Trans.(°↓) | | | PSNR | SSIM | LPIPS | Rot.(°↓) | | Trans.(°↓) | |
| | | | | Avg. | Med. | Avg. | Med. | | | | | Avg. | Med. | Avg. | Med. |
| MVSPlat | 23.993 | 0.784 | 0.154 | – | | | | | 23.003 | 0.777 | 0.203 | – | | | |
| CoPoNeRF | 16.138 | 0.427 | 0.483 | 8.778 | 2.791 | 24.036 | 18.432 | | 17.160 | 0.547 | 0.465 | 7.506 | 4.108 | 27.158 | 19.681 |
| Ours | **21.332** | **0.678** | **0.234** | **3.248** | **1.313** | **9.432** | **5.112** | | **21.877** | **0.733** | **0.221** | **2.778** | **1.511** | **12.881** | **7.882** |

truth target view image. For this evaluation, we compare our method with those of (Chen & Lee, 2023) and (Smith et al., 2023), since the method by (Hong et al., 2024) can only take two input views. We also report the Absolute Trajectory Error (ATE). The results are summarized in Tab. 5c. From these results, we find that our method achieves significantly better performance than the others, highlighting our capability to extend to multiple $N$ views. We also provide qualitative results and videos for $N$-view experiments, in supplementary material.

**Cross-Dataset Evaluation.** To demonstrate the generalization capability, we conduct a cross-dataset evaluation and compare against (Hong et al., 2024). Specifically, we evaluate performance on RealEstate10K and DL3DV, using each dataset for training in a cross-dataset setting. The results, summarized in Tab. 5d, show that our method achieves a PSNR of over 20 dB for both datasets, significantly outperforming (Hong et al., 2024). This indicates that, even under out-of-distribution conditions, our method produces high-quality renderings, highlighting its robustness and effectiveness in zero-shot capability.

## 5. Conclusion

In this paper, we have introduced learning-based framework that tackles pose-free novel view synthesis with 3DGS, enabling efficient, fast and photorealistic view synthesis from unposed images. Our framework, PFSplat, is built on foundation models to overcome inherent limitations of 3DGS. We have demonstrated that PF3plat is capable of training and inference solely from unposed images, even in scenarios where only a handful of images with minimal overlaps are given. Furthermore, PF3plat surpasses all existing methods on real-world large-scale datasets, establishing new state-of-the-art performance.

## Acknowledgement

This research was supported by Institute of Information & communications Technology Planning & Evaluation (IITP) grant funded by the Korea government (MSIT) (RS-2019-II190075, RS-2024-00509279, RS-2025-II212068, RS-2023-00227592, RS-2024-00457882) and the Culture, Sports, and Tourism R&D Program through the Korea Creative Content Agency grant funded by the Ministry of Culture, Sports and Tourism (RS-2024-00345025, RS-2024-00333068), and National Research Foundation of Korea (RS-2024-00346597).

## Impact Statement

This paper presents work whose goal is to advance the field of Machine Learning, which focuses in practical novel view synthesis. There are many potential societal consequences of our work, none of which we feel must be specifically highlighted here.

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

# A. Appendix

## A.1. Training Details

We train MVSplat (Chen et al., 2024) and CoPoNeRF (Hong et al., 2024) using our data loaders, similarly increasing the distance between context views during training, as explained in Sec 4.1. Specifically, we train MVSplat for 200,000 iterations using a batch size of 8 on a single A6000 GPU. All the hyperparameters are set to the authors' default setting. For CoPoNeRF, we train it for 50,000 iterations using 8 A6000 GPUs with effective batch size of 64, following the authors' original implementations and hyperparameters. Finally, for InstantSplat (Fan et al., 2024), we train and evaluate on a single A6000 GPU with a batch size of 1 by following the official code[1], and the hyperparameters were set according to the default settings provided by the authors.

## A.2. Implementation Details

We provide a detailed illustration in Fig. 2 for our fine alignment and confidence estimation modules.

## A.3. Evaluation Details

For the evaluation on RealEstate-10K and ACID, we follow the protocol outlined by (Hong et al., 2024), where evaluation is conducted using unposed triplet images $(I_1, I_2, I_t)$. The test set is divided into three groups, small, middle, and large, based on the extent of overlap between $I_1$ and $I_2$. This allows the model's performance to be assessed under varying levels of difficulty, reflecting different real-world scenarios. For the relatively new DL3DV dataset, we introduce a new evaluation protocol. For each scene, we select two context images, $I_1$ and $I_2$, by skipping frames at intervals of 5 and 10, creating two groups per scene, each representing small and large overlap cases. We then randomly select three target images from the sequence between the context images.

For evaluation metrics, we use standard image quality measures, PSNR, SSIM, LPIPS, and MSE, for novel view synthesis. For camera pose estimation, we compute the geodesic rotation error and angular difference in translation, as commonly done in classical methods (Nistér, 2004; Melekhov et al., 2017). Our statistical analysis reports both the average and median errors, with the median providing robustness against outliers.

## A.4. Additional Experiment

**Different Strategy for Coarse Alignment.** In this additional experiment, we analyze the impact of different coarse alignment strategies. Specifically, we replace LightGlue with RoMa (Edstedt et al., 2024) and UniDepth with Depth-

[1]https://github.com/NVlabs/InstantSplat

| Method | PSNR | SSIM | LPIPS | Rot. (°↓) | | Trans. (°↓) | |
|---|---|---|---|---|---|---|---|
| | | | | Avg. | Med. | Avg. | Med. |
| MASt3R | - | - | - | 2.555 | 0.751 | 9.775 | 2.830 |
| RoMa | - | - | - | 2.470 | 0.391 | 8.047 | 1.601 |
| Ours | 23.589 | 0.782 | 0.181 | 1.756 | 0.897 | 9.474 | 4.628 |
| DepthPro + Fine Align. (Ours) | 21.882 | 0.701 | 0.289 | 2.778 | 0.812 | 11.322 | 5.414 |
| RoMa + Fine Align. (Ours) | 24.412 | 0.799 | 0.167 | 2.152 | 0.501 | 7.544 | 3.233 |

*Table 6.* **Different Strategies for Coarse Alignment.**

Pro (Bochkovskii et al., 2024) for coarse alignment. The results, presented in Tab. 6, reveal that using more robust and accurate correspondences leads to noticeable improvements, as these correspondences enhance our loss computation. However, we observed that integrating DepthPro results in significant performance degradation. This issue may stem from the feature maps used by our fine alignment modules, which could potentially be mitigated through further engineering refinements.

## A.5. More Qualitative Results

Fig. 4, Fig. 5 and Fig. 6 present more novel view rendering results of different methods. Fig. 7 shows qualitative results on pose estimation in $N$-view experiments, compared against FlowCAM (Smith et al., 2023). Finally, we provide 6-view and 12-view qualitative results in Fig. 8 and Fig. 9, while videos can be found in the attached file.

## A.6. Limitations and Future Work

As our model currently lacks a mechanism to handle dynamic scenes, it is unable to accurately capture scene dynamics or perform view extrapolation. Additionally, our model's performance is contingent on the quality of the coarse alignments, which rely on the accuracy of the depth and correspondence models. In cases where either of these models fails, our approach may not function optimally. However, because our refinement modules are lightweight, simple, and model-agnostic, incorporating more advanced methods for coarse alignment could further enhance performance. Moreover, while existing pose-free view synthesis methods (Chen & Lee, 2023; Smith et al., 2023; Hong et al., 2024; Ye et al., 2024) share a limitation that requires camera intrinsic parameters, exploring joint optimization of both camera extrinsic and intrinsic is a promising direction. Finally, our method is not designed for dynamic scenes, but we believe exploring feed-forward dynamic scene reconstruction is a promising direction.

For future work, we plan to train our model on diverse large-scale datasets. Since our approach relies exclusively on supervision signals from RGB images, it is straightforward to scale up the training data. We also aim to extend our method to handle 4D objects, ultimately enabling the modeling of 4D scenes, which would be beneficial for applications such as egocentric vision and robotics.

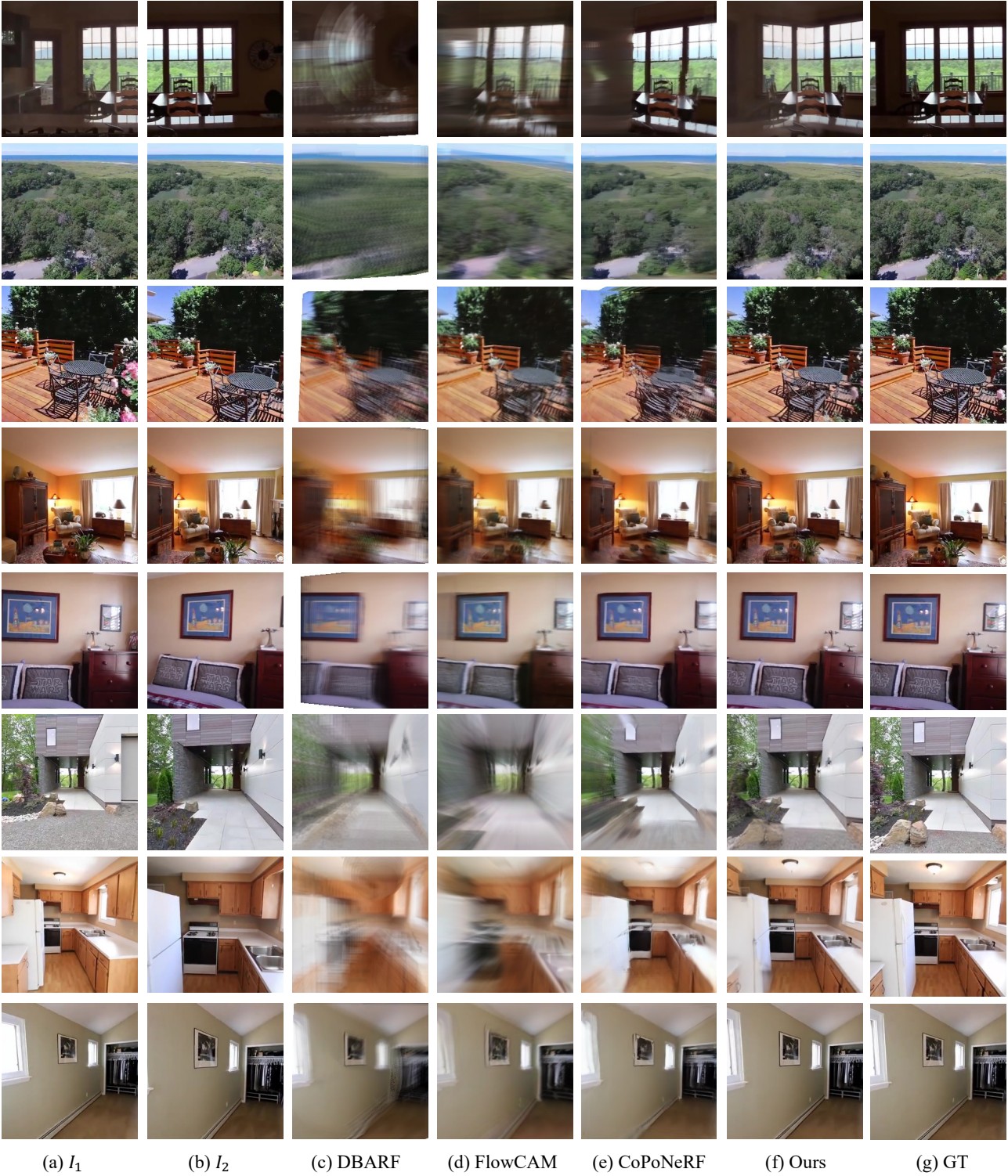

| (a) $I_1$ | (b) $I_2$ | (c) DBARF | (d) FlowCAM | (e) CoPoNeRF | (f) Ours | (g) GT |

*Figure 4.* **Qualitative results on RealEstate-10K dataset.** Given two context views (a) and (b), we compare novel view rendering results.

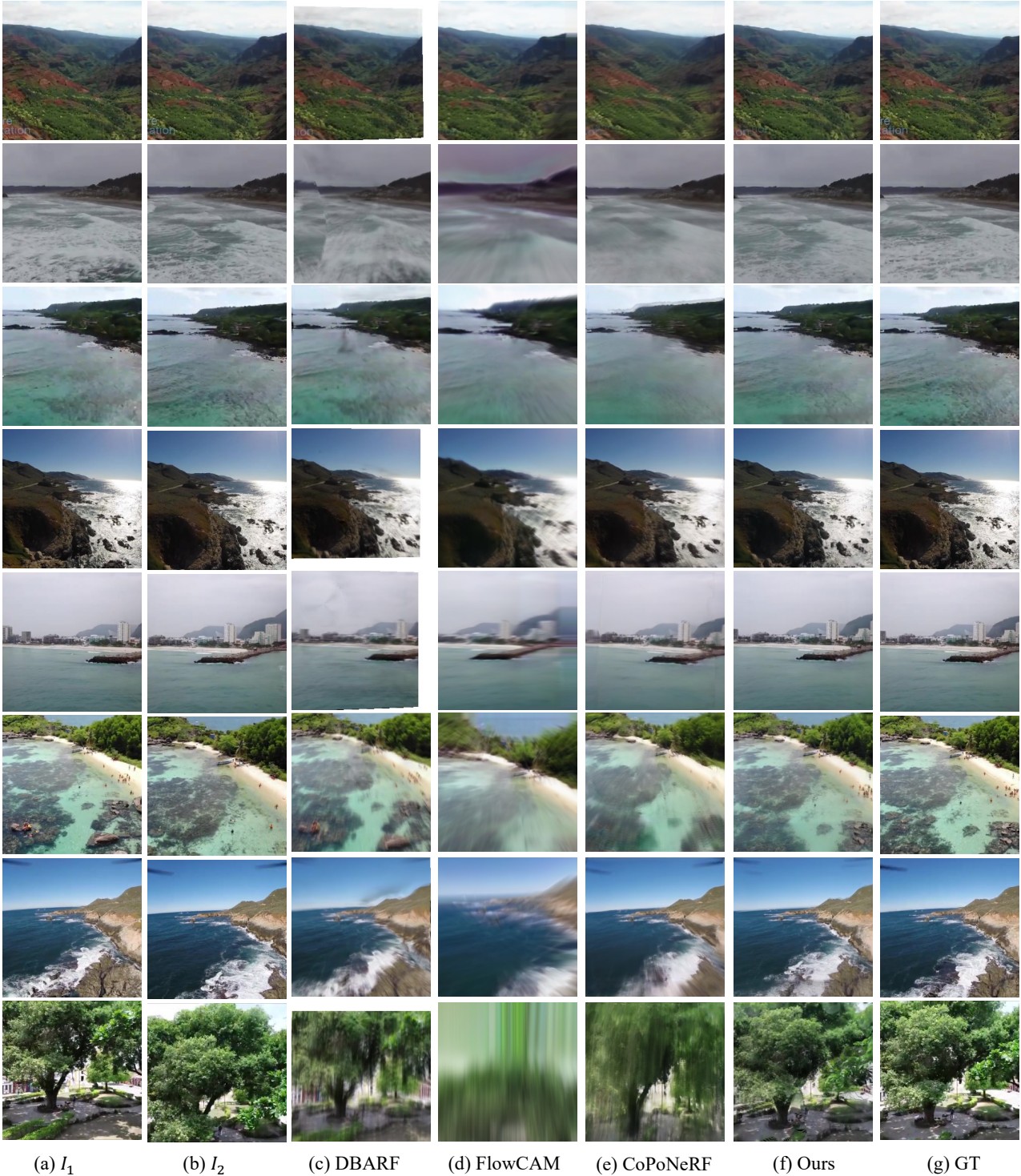

|  (a) $I_1$  |  (b) $I_2$  |  (c) DBARF  |  (d) FlowCAM  |  (e) CoPoNeRF  |  (f) Ours  |  (g) GT  |

*Figure 5.* **Qualitative results on ACID dataset.** Given two context views (a) and (b), we compare novel view rendering results.

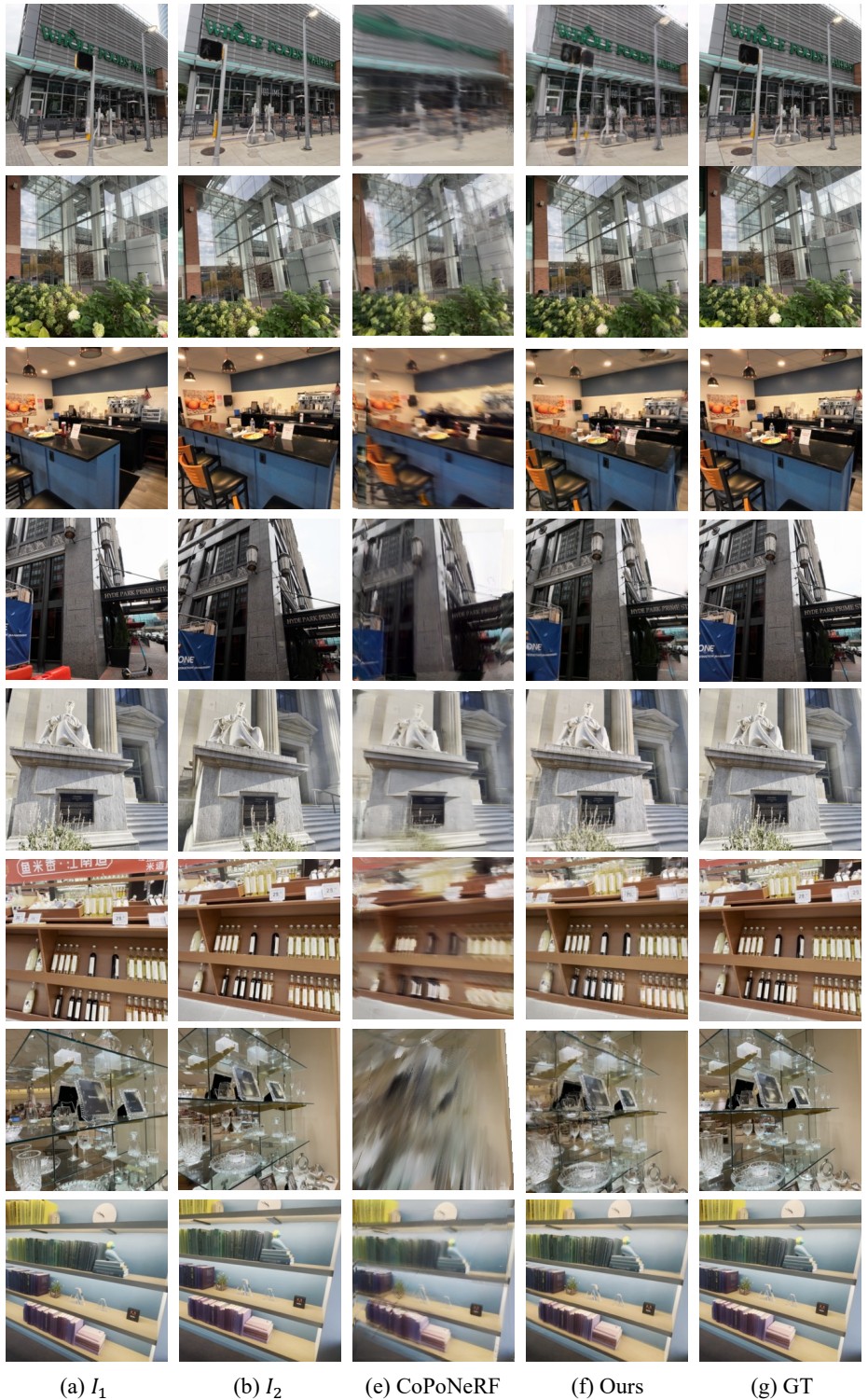

(a) $I_1$     (b) $I_2$     (e) CoPoNeRF     (f) Ours     (g) GT

*Figure 6.* **Qualitative results on DL3DV dataset.** Given two context views (a) and (b), we compare novel view rendering results.

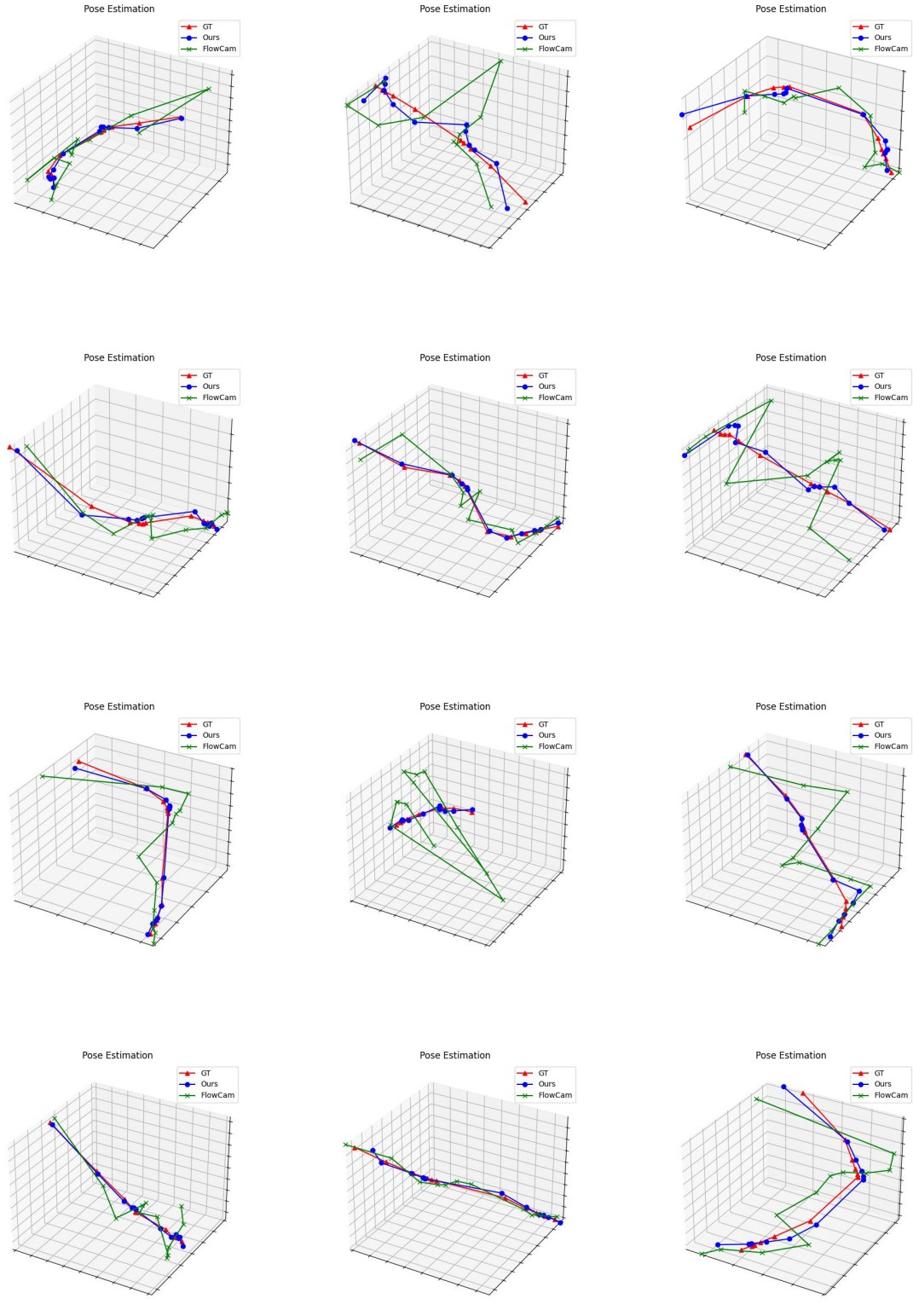

*Figure 7.* **Pose estimation qualitative results.**

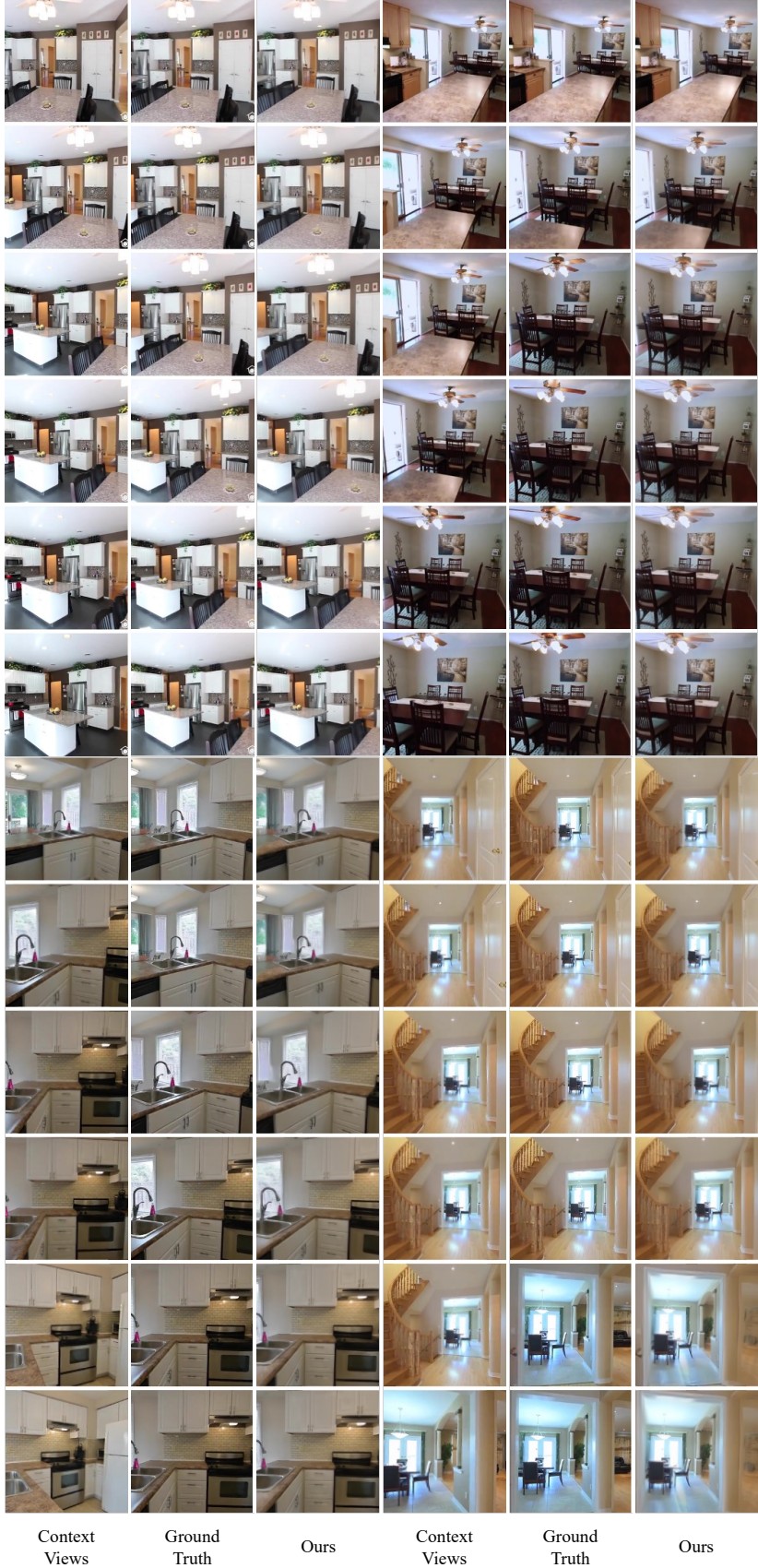

| Context Views | Ground Truth | Ours | Context Views | Ground Truth | Ours |

*Figure 8.* **Qualitative results on RealEstate-10K with 6 input views.**

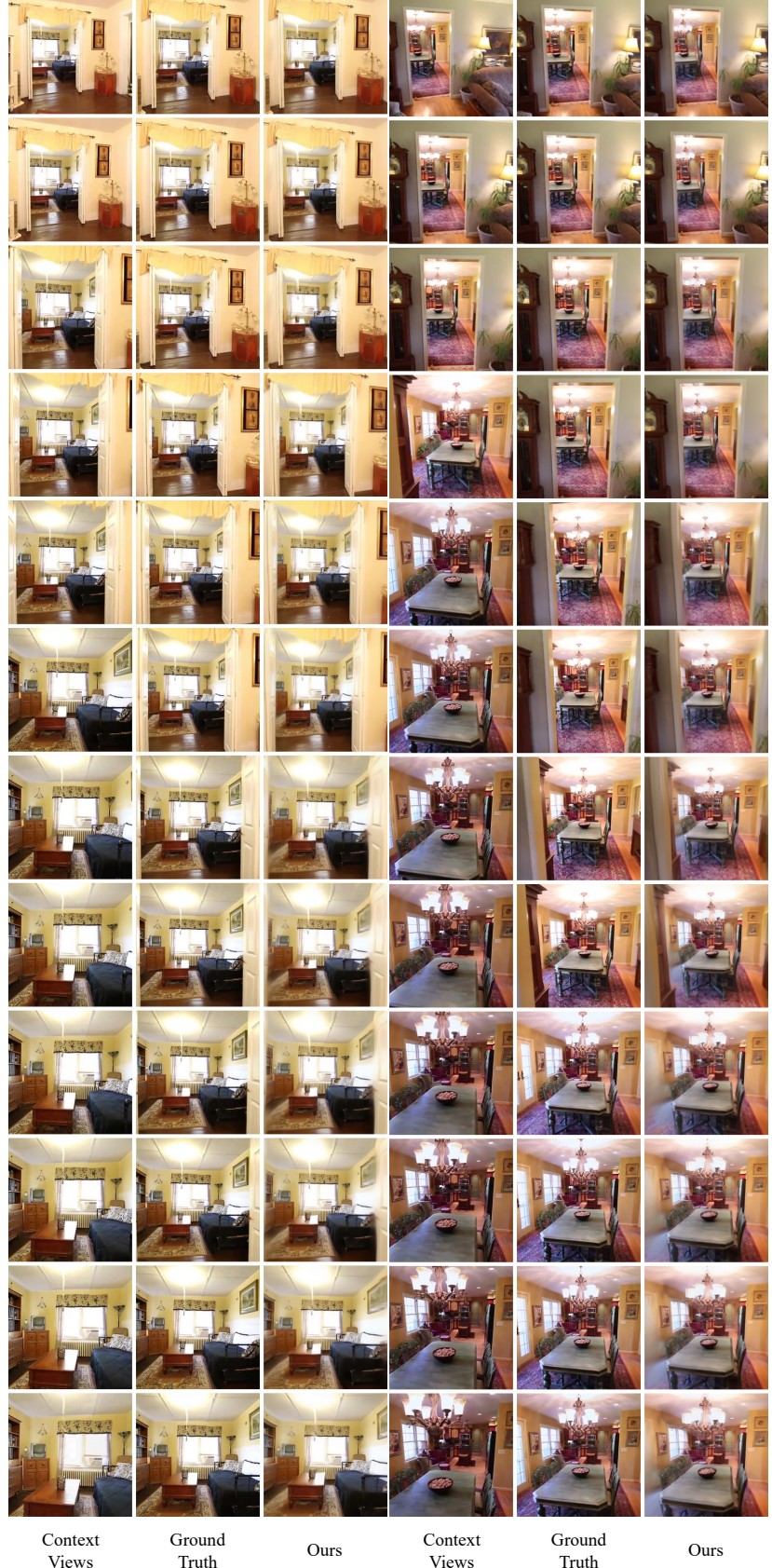

| Context Views | Ground Truth | Ours | Context Views | Ground Truth | Ours |

*Figure 9.* **Qualitative results on RealEstate-10K with 12 input views.**

