# OpenReview forum: "PF3plat: Pose-Free Feed-Forward 3D Gaussian Splatting for Novel View Synthesis"
_ICML.cc/2025/Conference — ICML 2025 poster_

### Official Review · Reviewer_Sfrw · 2025-02-26

**Overall Recommendation:** 3

**Summary:**

This paper introduces PF3Plat, a novel two-stage framework for novel view synthesis from sparse and unposed images. In the first stage, it leverages pre-trained depth estimation and visual correspondence models to achieve a coarse alignment of 3D Gaussians. In the second stage, it refines depth and pose estimation using geometry-aware modules.

## update after rebuttal

I remain positive due to the impressive experimental results.

**Claims And Evidence:**

The claims are well-supported by extensive experimental results.

**Essential References Not Discussed:**

To the best of my knowledge, none are missing.

**Experimental Designs Or Analyses:**

In Table 4, the baseline model is MVSplat with pose estimation from Stage 1. Why not use MVSplat with MASt3D, given that this method falls behind MASt3R in camera pose estimation?

**Methods And Evaluation Criteria:**

The paper presents a highly practical solution for novel view synthesis from unposed images.

**Other Comments Or Suggestions:**

None.

**Other Strengths And Weaknesses:**

This paper assumes known camera intrinsics, whereas DUSt3D does not have this requirement.

For camera pose estimation, this method underperforms compared to MASt3R, which does not require additional training on those datasets.

The improvement over InstantSplat is limited.

**Questions For Authors:**

None.

**Relation To Broader Scientific Literature:**

This paper is related to novel view synthesis, pose estimation, and 3D reconstruction.

**Theoretical Claims:**

To the best of my knowledge, the theoretical claims appear to be correct.

---

> ### Author Rebuttal · Authors · 2025-04-01
>
> > Why not use MVSplat with MASt3D, given that this method falls behind MASt3R in camera pose estimation?
>
> We opted not to use MVSplat with MASt3R because MASt3R's runtime of around 10 seconds conflicts with our goal of achieving a fast, feed-forward process.  Moreover, our experiments include a variant, MASt3R*, which employs PnP+RANSAC for pose estimation similar to our approach, yet MASt3R* performs worse than both our baseline and final model, as shown in Tab. 2. We believe these reasons justify our design choice for the baseline.
>
> Additionally, we wish to highlight that we have shown comparisons that outperform MASt3R. We kindly refer the reviewer to ***Tab. 6***, where RoMa outperforms MASt3R, and our variant that uses RoMa enables further improvements in both image and pose estimation quality. Through this experiment, we highlight the flexibility of model choice for coarse alignment stage and a potential for performance improvements.
>
> > This paper assumes known camera intrinsics, whereas DUSt3r does not have this requirement.
> We thank the reviewer for highlighting this point. As discussed in Sec. A.6, we acknowledge that assuming known camera intrinsics is a limitation common to many NVS methods. Nevertheless, we conducted an additional experiment using UniDepth's intrinsic predictions instead of ground truth, and the results indicate that our method still performs well without GT intrinsics.
>
>
> | Method                 | PSNR    | SSIM   | LPIPS |
> |------------------------|---------|--------|-------|
> | CoPoNeRF               | 19.536  |  0.638 | 0.398 |
> | Ours w/o GT intrinsics | 22.688  | 0.733  | 0.201 |
> | Ours                   | 23.589  | 0.782  | 0.181 |
>
> From the results, we highlight that even without GT intrinsics, ours  outperforms CoPoNeRF and also performs on par with ours with GT intrinsics. This suggests a potentially feasible approach to alleviate the limitation.
>
> Moreover, our primary focus is on generalized novel view synthesis from unposed images—where pose, depth, and correspondence estimation serve as intermediate tasks to achieve high-quality novel view rendering. In contrast to SfM-based methods such as DUSt3R, VGGSfM, and SP+SG, our approach differs in terms of data, objectives, and training setups. Notably, methods like DUSt3R and MASt3R require additional steps (e.g., training a radiance field via NeRF or 3D Gaussian Splatting), incur long optimization times per scene.
>
> > The improvement over InstantSplat is limited.
> We thank the reviewer for the comment. As shown in Tab. 5, our method outperforms InstantSplat by a large margin, and our inference speed is 100x faster, demonstrating a significant practical advantage. Moreover, when a more advanced correspondence network is adopted or test-time optimization is performed (like InstantSplat) for coarse alignment, training process or inference-phase optimization, our method further broadens the gap, as shown below:
>
> | Method       | PSNR   | SSIM  | LPIPS | Rot Avg | Rot Med | Trans Avg | Trans Med | Time  |
> |--------------|--------|-------|-------|---------|---------|-----------|-----------|-------|
> | InstantSplat | 23.079 | 0.777 | 0.182 | 2.693   | 0.882   | 11.866    | 3.094     | 53    |
> | Ours         | 23.589 | 0.782 | 0.181 | 1.756   | 0.897   | 9.474     | 4.628     | 0.390 |
> | Ours+TTO     | 24.689 | 0.798 | 0.167 | 1.662   | 0.871   | 8.998     | 4.311     | 24    |
> | Ours + RoMa  | 24.412 | 0.799 | 0.167 | 2.152   | 0.501   | 7.544     | 3.233     | 0.523 |

---

> > ### Comment · Reviewer_Sfrw · 2025-04-04
> >
> > Thank you for addressing my questions. Based on the its efficiency and performance on novel view synthesis, i am leaning toward accepting this paper.

---

> > > ### Author Response · Authors · 2025-04-05
> > >
> > > We thank the reviewer for acknowledging our rebuttal and for the appreciation of our work. If we have successfully addressed all of your concerns, we would be highly grateful if the reviewer could consider increasing the rating, as it would significantly support our submission.

---

### Official Review · Reviewer_iLyu · 2025-03-10

**Overall Recommendation:** 2

**Summary:**

This paper proposes a novel 3D Gaussian Splatting prediction model based on sparse views. Starting from a coarse initialization using off-the-shelf depth and correspondence models, the proposed fine alignment module predicts the correct scale for the depth map and camera poses, and finally, Gaussian heads predict the 3DGS parameters. The results demonstrate improved performance over baseline methods.

**Claims And Evidence:**

There are several concerns regarding the claim:
- I do not think the proposed method qualifies as a “feed-forward” approach, because the coarse initialization stage employs a test-time iterative optimization for initial pose estimation. A feed-forward model typically involves directly regressing scene parameters in an end-to-end manner, but the proposed method is more like an engineered system that combines several off-the-shelf pre-trained models and iterative optimization modules, raising questions about its elegance and novelty.
- The authors assume that camera intrinsics are generally available, which is not always the case for generic video analysis tasks.

**Essential References Not Discussed:**

No.

**Experimental Designs Or Analyses:**

Checked.

**Methods And Evaluation Criteria:**

The benchmark datasets are reasonable; however, the study overlooks some state-of-the-art baselines, such as NoPoSplat and MASt3R-SfM, both of which had publicly available code well before the ICML submission deadline.

**Other Comments Or Suggestions:**

No

**Other Strengths And Weaknesses:**

While the method demonstrates a practical system design and solid performance on benchmark datasets, it appears to be a composite system that lacks significant scientific insight.

**Questions For Authors:**

Already commented questions and concerns.

**Relation To Broader Scientific Literature:**

As noted in the "Methods and Evaluation Criteria" section, some important baselines are missing.

**Theoretical Claims:**

Checked.

---

> ### Author Rebuttal · Authors · 2025-04-01
>
> > Questions about “feed-forward” approach, because there is a module for a test-time iterative optimization for initial pose.
>
> We wish to refer to reviewer k81v’s appreciation that ***“feed-forward” in our context emphasizes efficiency and speed*** (See comparison to InstantSplat).  Although our coarse initialization stage employs iterative optimization, specifically, PnP+RANSAC, which is recognized as fast and efficient that takes only milliseconds (as also noted in NoPoSplat), the subsequent fine alignment module is fully feed-forward and represents a novel contribution. This fine alignment refines the coarse pose to correct 3D Gaussian misalignments, leading to significant performance gains (see Tab. 4).  Leveraging off-the-shelf models for coarse alignment allowed us to focus on this key challenge (Our novelty in this aspect of coarse alignment is also recognized by the reviewer TAPU and EyLp),  and we believe that our framework is novel and addresses a very important task in computer vision, which are appreciated by reviewer K81V and EyLp. We hope that this minor terminology point should not impact the overall evaluation of our work.
>
> > The authors assume that camera intrinsics are generally available, which is not always the case for generic video analysis tasks.
>
> While we acknowledge that there are some corner cases where intrinsics might not be available, ***this limitation is common to many pose-free feed-forward NVS methods, as discussed in Sec. A.6.***  Nonetheless, when using intrinsic estimates from UniDepth at inference, our method still outperforms CoPoNeRF as shown below. We wish to stress that in practical applications, intrinsics are generally available, and even in cases where they are not, our approach maintains competitive performance relative to other methods facing the same limitation.
>
>
> | Method                 | PSNR    | SSIM   | LPIPS |
> |------------------------|---------|--------|-------|
> | CoPoNeRF               | 19.536  |  0.638 | 0.398 |
> | Ours w/o GT intrinsics | 22.688  | 0.733  | 0.201 |
> | Ours                   | 23.589  | 0.782  | 0.181 |
>
> From the results, we highlight that even without GT intrinsics, ours outperforms CoPoNeRF and also performs on par with ours with GT intrinsics. This suggests a potentially feasible approach to alleviate the limitation.
>
> > The study overlooks some state-of-the-art baselines, such as NoPoSplat and MASt3R-SfM, both of which had publicly available code well before the ICML submission deadline.
>
> We thank the reviewer for highlighting NoPoSplat and MASt3R-SfM. As noted by reviewer EyLp and the reviewer guideline explicitly states, ***NoPoSplat is a concurrent work—released on arXiv less than three months before our ICML submission and accepted by ICLR three days prior—so we are not obliged to compare against it***, though it is acknowledged in the related works (Nonetheless, we provide additional comparison below, as the reviewer kindly suggests). Moreover, MASt3R-SfM is  an arXiv paper and, while it shares similar objectives in 3D reconstruction, it addresses a different task than our pose-free feed-forward NVS (its sparse view variants (MASt3R) are already compared with our method in Tab. 2).
>
>  Below includes the comparison:
>
> | Methods | Pose Supervision | PSNR $\uparrow$ | LPIPS $\downarrow$ | SSIM $\uparrow$  |
> | --- | :---: | --- | --- | --- |
> | CoPoNeRF | $\checkmark$ | 19.536 | 0.398 | 0.638 |
> | NoPoSplat | $\checkmark$ | 26.820 | 0.126 | 0.879 |
> | DBARF | $\times$ | 14.789 | 0.490 | 0.570 |
> | FlowCam | $\times$ | 18.242 | 0.597 | 0.455 |
> | Ours | $\times$ | 23.589 | 0.181 | 0.782 |
>
> While NoPoSplat performs well, as indicated in the table, we stress that our method clearly differentiates in the setting that the 3D geometry data, such as GT camera pose, is not utilized during training. Moreover, we show in the following that our method can be naturally extended to N-view setting, where NoPoSplat struggles.
>
> | Methods (6 views) | Pose Supervision | PSNR $\uparrow$ | LPIPS $\downarrow$ | SSIM $\uparrow$  |
> | --- | :---: | --- | --- | --- |
> | NoPoSplat | $\checkmark$ | 18.007 | 0.384 | 0.584 |
> | Ours | $\times$ | 27.028 | 0.116 | 0.879 |
>
> | Methods (12 views) | Pose Supervision | PSNR $\uparrow$ | LPIPS $\downarrow$ | SSIM $\uparrow$  |
> | --- | :---: | --- | --- | --- |
> | NoPoSplat | $\checkmark$ | 17.625 | 0.399 | 0.583 |
> | Ours | $\times$ | 28.133 | 0.099 | 0.993 |
>
> To summarize, compared to NoPoSplat, our approach offers key advantages compared to NoPoSplat: it is a conceptual contribution proposing a general pipeline for pose-free view synthesis that addresses 3D Gaussian misalignment using off-the-shelf depth and correspondence networks and fine alignment modules; it naturally extends to N-view inference without retraining; and it is designed for scenarios without 3D geometry data (GT camera pose) during training. We will additionally include this discussion in related works.

---

### Official Review · Reviewer_K81V · 2025-03-12

**Overall Recommendation:** 3

**Summary:**

The paper introduces PF3plat to address the problem pose-free feed-forward view synthesis using 3D Gaussians parametrization. First, the method leverages pretrained monocular depth estimator and visual correspondence models to get coarse depth and pose. Then learnable refinement modules are proposed to refine the depth and pose, which are conditioned on estimated confidence using cost volumes. The method further utilizes the 2D-3D and 3D-3D consistent loss to regularize the geometry. It achieves sota performance in pose-free synthesis methods.

## update after rebuttal
The contribution of work mainly relies on the refinement module and new losses, which I argree is incremental rather than substantial. I also encourage the authors to include more analysis/qualitative experiments of the ablation studies to demonstrate their proposed method better. Given the good empirical performance, I decided to maintain my score.

**Claims And Evidence:**

Claims are supported by qualitative and quantative results of novel view synthesis and pose estimation on RealEstate10K, ACID, and DL3DV datasets.

**Essential References Not Discussed:**

Another pose-free feed-forward Gaussian Splatting paper FreeSplatter[1] not discussed in the paper.

[1] Xu, Jiale, Shenghua Gao, and Ying Shan. "FreeSplatter: Pose-free Gaussian Splatting for Sparse-view 3D Reconstruction." arXiv preprint arXiv:2412.09573 (2024).

**Experimental Designs Or Analyses:**

1. Experimental designs on unposed triplet, with the test set divided into small, middle, and large based on overlap between input views follows .
2. Separating the pose-free and pose-required methods in results is resonable.
3. Experiments and analyses in comparison to scene-optimization approach, speed, and cross-domain results are valid.

**Methods And Evaluation Criteria:**

Utilizing pretrained models and refining them to predict 3D Gaussians makes sense. Evaluation includes both indoor and outdoor real-world datasets, following pixelSplat[1] , MVSplat[2], and NoPoSplat[3].

[1].Charatan, D., Li, S., Tagliasacchi, A., and Sitzmann, V. pixelsplat: 3d gaussian splats from image pairs for scalable generalizable 3d reconstruction. arXiv:2312.12337, 2023.arXiv preprint
[2].Chen, Y., Xu, H., Zheng, C., Zhuang, B., Pollefeys, M., Geiger, A., Cham, T.-J., and Cai, J. Mvsplat: Efficient 3d gaussian splatting from sparse multi-view images. arXiv preprint arXiv:2403.14627, 2024.
[3].Ye, B., Liu, S., Xu, H., Li, X., Pollefeys, M., Yang, M.-H., and Peng, S. No pose, no problem: Surprisingly simple 3d gaussian splats from sparse unposed images. arXiv preprint arXiv:2410.24207, 2024.

**Other Comments Or Suggestions:**

1. In Table 4, I assume (I-III) and (I-IV) are 2D-3D and 3D-3D consistent loss. The naming are somewhat confusing. These losses are essential to achieve good quality based on the results. In the current writing, however, they are not introduced in the earlier part of the paper.
2. In Table 4, Please explain (I-II) Scale/Shift Tuning Depth Network in the paper.
3. There is a typo "sprase" in line 154.

**Other Strengths And Weaknesses:**

Strengths
1. The task is relevant and important in the field, and the paper is overall well-written and easy to follow.
2. The proposed coarse and refinement pipeline together with other design choices mentioned above is effective and outperform other pose-free methods.

Weaknesses
1. The contribution mainly relies on the refinement modules. While it is effective, much of the work builds on existing works or ideas, such as Unidepth[1], LightGlue[2], cost volumes[3].
2. NoPoSplat [4] is acknowledged but not evaluated and compared.
3. More qualitative analysis is expected for the ablation study.

[1]. Piccinelli, L., Yang, Y.-H., Sakaridis, C., Segu, M., Li, S., Van Gool, L., and Yu, F. Unidepth: Universal monoc- ular metric depth estimation. In Proceedings of the IEEE/CVF Conference on Computer Vision and Pattern Recognition, pp. 10106–10116, 2024
[2]. Lindenberger, P., Sarlin, P.-E., and Pollefeys, M. Lightglue: Local feature matching at light speed. In Proceedings of the IEEE/CVF International Conference on Computer Vision, pp. 17627–17638, 2023.
[3]. Chen, Y., Xu, H., Zheng, C., Zhuang, B., Pollefeys, M., Geiger, A., Cham, T.-J., and Cai, J. Mvsplat: Efficient 3d gaussian splatting from sparse multi-view images. arXiv preprint arXiv:2403.14627, 2024.
[4]. Ye, B., Liu, S., Xu, H., Li, X., Pollefeys, M., Yang, M.-H., and Peng, S. No pose, no problem: Surprisingly simple 3d gaussian splats from sparse unposed images. arXiv preprint arXiv:2410.24207, 2024.

**Questions For Authors:**

1. Around line 211 and 212, does the inputs to 3D Gaussian parameter prediction also include the refined pose? I assume that gradient can also be flowed from the refined depth input. Is this the case?
2. Followed by question 1, could you explain more about the intuitation of confidence estimation and what exactly does it help with the reconstruction overall?
3. How did you determine the lambda for 3D-3D loss?

**Relation To Broader Scientific Literature:**

The key contributions of the paper relate to how to leverage pretrained models and refine their estimations to predict 3D structures. Certain design choices, e.g. network architecture, cost volumes, training losses contribute the performance. The pose-free setting can be applied to in-the-wild scenarios and general 3D reconstruction. Feed-forward manner relates to the efficiency and speed of such methods.

**Theoretical Claims:**

In the last equation in equation 2, t misses subscript i. The architecture of T_agg is not specified.

---

> ### Author Rebuttal · Authors · 2025-04-01
>
> > FreeSplatter not discussed in the paper.
>
> We thank the reviewer for highlighting FreeSplatter[1]. FreeSplatter addresses the same task using an LRM-based architecture that directly maps images to 3D Gaussians. To tackle coarse alignment, it employs staged training with early supervision from 3D geometry data (ground-truth pointmaps constructed using pose and depth), which is essential for convergence. This supports our claim that misaligned 3D Gaussians hinder learning, and contrasts with our approach, which incorporates a dedicated coarse alignment module. We will include this discussion in the related works section.
>
> > The contribution mainly relies on the refinement modules. While it is effective, much of the work builds on existing works or ideas.
>
> While our approach leverages existing methods for coarse alignment (while recognized as creative by reviewer TAPU), our main technical contributions lie in the novel fine-alignment modules. In addition to our findings to identify a unique challenge in pose-free feed-forward 3DGS, we address the challenge by proposing new losses, fine alignment modules, and a confidence estimation module. ***As demonstrated in Table 4, each component significantly contributes to performance improvements***, underscoring the technical innovation of our work.
>
> > NoPoSplat [4] is acknowledged but not evaluated and compared.
>
> As mentioned by the reviewer EyLp, ***NoPoSplat is a concurrent work released on arXiv less than three months ago and accepted only three days before our submission.*** Moreover, as shown in Table 2, fair comparison is challenging since other works (CoPoNeRF and NoPoSplat) use additional supervisory data (GT camera pose) during training. Nonetheless, we agree that including NoPoSplat as a baseline would further enhance our work.  Below includes the comparison:
>
> | Methods | Pose Supervision | PSNR $\uparrow$ | LPIPS $\downarrow$ | SSIM $\uparrow$  |
> | --- | :---: | --- | --- | --- |
> | CoPoNeRF | $\checkmark$ | 19.536 | 0.398 | 0.638 |
> | NoPoSplat | $\checkmark$ | 26.820 | 0.126 | 0.879 |
> | DBARF | $\times$ | 14.789 | 0.490 | 0.570 |
> | FlowCam | $\times$ | 18.242 | 0.597 | 0.455 |
> | Ours | $\times$ | 23.589 | 0.181 | 0.782 |
>
> While NoPoSplat performs well, as indicated in the table, we stress that our method clearly differentiates as they use GT camera pose during training. Moreover, below shows our method can be naturally extended to n-view setting, where NoPoSplat struggles.
>
> | Methods (6 views) | Pose Supervision | PSNR $\uparrow$ | LPIPS $\downarrow$ | SSIM $\uparrow$  |
> | --- | :---: | --- | --- | --- |
> | NoPoSplat | $\checkmark$ | 18.007 | 0.384 | 0.584 |
> | Ours | $\times$ | 27.028 | 0.116 | 0.879 |
>
> | Methods (12 views) | Pose Supervision | PSNR $\uparrow$ | LPIPS $\downarrow$ | SSIM $\uparrow$  |
> | --- | :---: | --- | --- | --- |
> | NoPoSplat | $\checkmark$ | 17.625 | 0.399 | 0.583 |
> | Ours | $\times$ | 28.133 | 0.099 | 0.993 |
>
> > More qualitative analysis for the ablation study.
>
> We thank the reviewer for the valuable suggestion. Although it has been challenging to identify samples that clearly distinguish all the variants in Table 4 during rebuttal period, we agree that including such examples would further strengthen our work. We will incorporate additional qualitative results in the updated version of the paper.
>
> > In Table 4, Please explain (I-II) Scale/Shift Tuning Depth Network in the paper.
>
> Variant (I-II) in Table 4 refers to the approach where we fine-tune the scale and shift parameters used to convert relative depth estimates into UniDepth predictions. We observed that directly fine-tuning the UniDepth parameters resulted in unstable training, indicating that a more sophisticated training strategy may be required.
>
> > Around line 211 and 212, does the inputs to 3D Gaussian parameter prediction also include the refined pose? I assume that gradient can also be flowed from the refined depth input.
>
> Yes, that's correct. The refined pose is used to construct the standard MVS cost volume, which is then aggregated with the guidance cost volume. Consequently, gradients do flow from the refined depth input as well, contributing to the overall optimization process.
>
> > Intuition of confidence estimation and what exactly does it help with the reconstruction overall?
>
> This confidence score is used to modulate the regression of 3D Gaussian parameters, such as opacities and covariances, ensuring that more reliable matches have a greater influence on the reconstruction. Empirically, this approach improves both rendering quality and pose estimation accuracy, leading to an overall improved reconstruction, as shown in the Tab. 4.
>
> > How did you determine the lambda for 3D-3D loss?
>
> We set the lambda for the 3D-3D loss to 0.05 by balancing its scale with the other losses in our framework. This value was chosen empirically, and a more extensive hyperparameter search might yield performance improvements.

---

### Official Review · Reviewer_EyLp · 2025-03-15

**Overall Recommendation:** 3

**Summary:**

The paper presents a novel feed-forward method for 3D reconstruction and view synthesis from sparse, unposed images, eliminating the need for ground-truth depth or pose at both training and inference. The method builds on pixel-aligned 3D Gaussian Splatting (3DGS) but addresses the challenge of Gaussian center misalignment, which can destabilize training.
To tackle this, the method first leverages off-the-shelf monocular depth estimation and image correspondence models to infer coarse depth and camera pose estimates. It then refines these estimates through a multi-view refinement process using learned modules, improving reconstruction quality and stability. Finally, the method computes geometry-aware confidence scores to assess the reliability of Gaussian centers, which condition the prediction of opacity, covariance, and color.
Using large-scale real-world indoor and outdoor datasets, the paper demonstrates that this method outperforms existing approaches in both rendered image quality and inferred camera pose accuracy, setting a new state-of-the-art in pose-free generalizable novel view synthesis.

## update after rebuttal
After reading the rebuttal, I think the paper makes a valuable contribution to pose-free novel view synthesis. The comparison to NoPoSplat is reasonable—it’s a concurrent paper, and the authors added results in the rebuttal showing that their method is competitive, faster, and works in more general settings. They also compared to MASt3R + MVSplat and showed better results and faster runtime.
I still recommend weak accept, and I believe the paper is above the bar.

**Claims And Evidence:**

The authors’ claim of state-of-the-art pose-free generalizable novel view synthesis is supported by extensive quantitative and qualitative analysis. See more in “Evaluation Criteria.”
Their claim that improved depth and camera pose estimates enhance pixel-aligned 3D Gaussian Splatting is backed by their ablation analysis, which demonstrates that refining scene estimates contributes to better view synthesis quality.
Additionally, their experimental results suggest that proper initialization is critical for stable training, as training without it leads to convergence issues. However, a deeper theoretical discussion on why inaccuracies in 3D Gaussian center localization cause noisy and sparse gradients—and whether alternative initialization strategies could mitigate this—would further strengthen their argument (see "Theoretical Claims").

**Essential References Not Discussed:**

I did not notice any missing citations. However, I recommend including works that further analyze the sensitivity of 3D Gaussian Splatting (3DGS) synthesis quality to the initial locations of the 3D Gaussian centers, which would provide valuable context and support on the matter.

**Experimental Designs Or Analyses:**

The experiments are well-structured and include strong ablations on different model components.

**Methods And Evaluation Criteria:**

The authors evaluate their method on two tasks: novel-view synthesis and camera pose estimation. Their model is trained and tested on three large-scale datasets: RealEstate10K (Zhou et al., 2018), ACID (Liu et al., 2021), and DL3DV (Lang et al., 2024).

For novel-view synthesis, they use standard image quality metrics, including PSNR, SSIM, LPIPS, and MSE. Camera pose estimation is assessed using the geodesic rotation error and angular difference in translation. This evaluation protocol follows a pose-free sparse view reconstruction method (Hong et al.) and has been adopted by other works in the field.

While the authors acknowledge NoPoSplat (Ye et al., 2024), a concurrent work achieving state-of-the-art results in pose-free generalizable novel-view synthesis from two views, they do not include it as a baseline for comparison. Given that NoPoSplat was accepted to ICLR 2025 only 50 days before the submission deadline, its omission is understandable. However, incorporating it as a baseline in future revisions would further strengthen the claims of state-of-the-art performance.

**Other Comments Or Suggestions:**

In Introduction (line 27, right column) unnecessary repetition of citation.

PF3plat is sometimes written as “PFsplat”.

**Other Strengths And Weaknesses:**

The claim and demonstration that increasing metric depth and camera pose inference to a sufficiently high level is enough to infer the locations of Gaussian centers for high-quality view synthesis is original and presents an intriguing direction for future work.

In addition, multiview refinement of monocular predictions is an interesting direction on its own, with potential of surpassing sparse view depth estimators.

Most sections of the paper are written clearly. The authors include and describe all relevant literature and related works effectively. The evaluation protocol follows established standards in the field and is well-described.


However, the description of the method could be clearer. It takes some time to understand that the Gaussian centers are derived directly from the depth of the pixels, rather than from a learned module. A more refined overview of the method, along with a revision of Figure 1 to better explain how and when each Gaussian parameter is estimated, could improve clarity.

**Questions For Authors:**

What is the performance of NoPoSplat in comparison to your approach?

Did you consider or experiment with other parameterization strategies or approaches to mitigate the impact of inaccuracies in 3D Gaussian center localization on noisy and sparse gradients?

Could you provide further analysis on why inaccuracies in 3D Gaussian center localization result in noisy and sparse gradients? Specifically, what factors contribute to this phenomenon?

**Relation To Broader Scientific Literature:**

The paper presents a novel feed-forward method for 3D reconstruction and view synthesis from sparse, unposed images, situating itself within the existing research. These methods open the door to applicability in real-world settings, where casually captured images contain sparse and distant viewpoints, and lack precise camera poses.

**Theoretical Claims:**

The authors do not provide any formal proofs. However, their hypothesis that inaccuracies in the localization of 3D Gaussian centers lead to noisy and sparse gradients, which cannot be easily compensated for, is supported by empirical evidence. Specifically, their results show that training without proper initialization leads to almost intractable convergence, reinforcing this claim.
While their findings provide some support, a more thorough analysis of the underlying causes could further strengthen their argument. Including citations that discuss this phenomenon would also improve credibility. Additionally, if training without initialization leads to intractable results, it would be valuable to explore whether alternative initialization strategies—beyond depth-based alignment—could be viable.
Prior work, such as PixelSplat, demonstrated that parameterizing Gaussian positions implicitly via dense probability distributions can mitigate local minima issues when optimizing primitive parameters through gradient descent. This suggests a potential complementary approach that the authors could explore. Integrating implicit parameterization into their method might further improve robustness.
In addition, the loss functions are mathematically well-defined.

---

> ### Author Rebuttal · Authors · 2025-04-01
>
> > Given that NoPoSplat was accepted to ICLR 2025 only few days before the ICML submission deadline, its omission is understandable. However, incorporating it as a baseline in future revisions would further strengthen the claims of state-of-the-art performance.
>
> We appreciate the reviewer’s valuable suggestion. As noted, NoPoSplat is a concurrent work released on arXiv less than three months ago and accepted only three days before our submission. Moreover, as indicated in Table 2, fair comparison is challenging since other works (CoPoNeRF and NoPoSplat) use additional supervisory data (GT Camera pose) during training. In contrast, our approach can be learned without such data, clearly differentiating our approach to them. Nonetheless, we agree that including NoPoSplat as a baseline along with detailed comparison would further enhance our work.  Below includes the comparison:
>
> | Methods | Pose Supervision | PSNR $\uparrow$ | LPIPS $\downarrow$ | SSIM $\uparrow$  |
> | --- | :---: | --- | --- | --- |
> | CoPoNeRF | $\checkmark$ | 19.536 | 0.398 | 0.638 |
> | NoPoSplat | $\checkmark$ | 26.820 | 0.126 | 0.879 |
> | DBARF | $\times$ | 14.789 | 0.490 | 0.570 |
> | FlowCam | $\times$ | 18.242 | 0.597 | 0.455 |
> | Ours | $\times$ | 23.589 | 0.181 | 0.782 |
>
>
> While NoPoSplat performs well, as indicated in the table, we stress that our method clearly differentiates in the setting that the 3D geometry data, such as GT camera pose, is not utilized during training. Moreover, we show in the following that our method can be naturally extended to N-view setting, where NoPoSplat struggles.
>
> | Methods (6 views) | Pose Supervision | PSNR $\uparrow$ | LPIPS $\downarrow$ | SSIM $\uparrow$  |
> | --- | :---: | --- | --- | --- |
> | NoPoSplat | $\checkmark$ | 18.007 | 0.384 | 0.584 |
> | Ours | $\times$ | 27.028 | 0.116 | 0.879 |
>
> | Methods (12 views) | Pose Supervision | PSNR $\uparrow$ | LPIPS $\downarrow$ | SSIM $\uparrow$  |
> | --- | :---: | --- | --- | --- |
> | NoPoSplat | $\checkmark$ | 17.625 | 0.399 | 0.583 |
> | Ours | $\times$ | 28.133 | 0.099 | 0.993 |
>
> To summarize, compared to NoPoSplat, our approach offers key advantages compared to NoPoSplat: it is a conceptual contribution proposing a general pipeline for pose-free view synthesis that addresses 3D Gaussian misalignment using off-the-shelf depth and correspondence networks and fine alignment modules; it naturally extends to N-view inference without retraining; and it is designed for scenarios without 3D geometry data (GT camera pose) during training. We will additionally include this discussion in related works.
>
> > Did you consider or experiment with other parameterization strategies or approaches to mitigate the impact of inaccuracies in 3D Gaussian center localization on noisy and sparse gradients?
>
> We appreciate the reviewer’s suggestion. While we have not yet experimented with alternative strategies like PixelSplat, we agree that mitigating inaccuracies in 3D Gaussian center localization is promising. Future work could explore leveraging NeRF representations, as in RADSplat, or formulating a probabilistic approach similar to PixelSplat. We will include a discussion of these alternatives in the supplementary material.
>
> > Could you provide further analysis on why inaccuracies in 3D Gaussian center localization result in noisy and sparse gradients? Specifically, what factors contribute to this phenomenon? Including citations that discuss this phenomenon would also improve credibility.
>
> We thank the reviewer for the valuable suggestion. In our framework, we claim that inaccuracies in 3D Gaussian center localization lead to noisy and sparse gradients because only the Gaussians within the rasterization window receive gradients. When centers are mislocalized, few Gaussians get effective updates, resulting in weak guidance, especially when initialized far from the optimal positions [1]. For this, we stressed the importance of coarse alignment. Moreover, providing photometric only showed limited effectiveness, in which we proposed 2D-3D consistency and regularization losses for addressing this.  Additionally, [2] and NoPoSplat highlight that when the network is initialized without 3D priors and reliable 3D geometry data during early training (e.g., GT pointmaps) to compensate for such priors is absent, convergence is difficult (Early training stage for FreeSplatter and CroCo initialization for NoPoSplat). We will include these discussions and citations in the final version of the paper.
>
>
> [1] Jung, J., Han, J., An, H., Kang, J., Park, S. and Kim, S., 2024. Relaxing accurate initialization constraint for 3d gaussian splatting. arXiv preprint arXiv:2403.09413.
>
> [2] Xu, Jiale, Shenghua Gao, and Ying Shan. "FreeSplatter: Pose-free Gaussian Splatting for Sparse-view 3D Reconstruction." arXiv preprint arXiv:2412.09573 (2024).

---

### Official Review · Reviewer_TAPU · 2025-03-23

**Overall Recommendation:** 2

**Summary:**

This work presents a framework for novel view synthesis (NVS) from unposed images in a single feed-forward pass. It estimates depth and pose from unposed images using a combination of pre-trained monocular depth estimation and visual correspondence models. It outperforms prior pose-free methods like DBARF and CoPoNeRF in NVS quality and pose estimation.

**Claims And Evidence:**

The claims in the paper are generally supported by experimental results, ablation studies, and comparisons with prior work.
1) The method shows clear improvements over prior pose-free methods like DBARF and CoPoNeRF in PSNR, SSIM, and LPIPS metrics on RealEstate-10K and ACID.
2) The ablation studies in Table 4 demonstrate a clear drop in performance when removing depth refinement, pose refinement, or geometry-aware confidence scores. They effectively validate the effect of these components.
3) The inference time of 0.39s per view (Table 5a) is significantly faster than InstantSplat (53s), making PF3plat more practical.

However, some claims could be better substantiated with additional experiments.
a) "Our method improves robustness in regions with low texture or significant viewpoint changes." The paper states that correspondence models like LightGlue struggle in low-texture regions. It is unclear how PF3plat can improve in these areas.
b) Lack of qualitative results for pose estimation. While the paper presents quantitative results (e.g., rotation and translation errors), it does not provide visual examples of estimated vs. ground truth poses, which would offer clearer insight into the actual performance of the pose refinement module.

**Essential References Not Discussed:**

N/A

**Experimental Designs Or Analyses:**

Yes, I reviewed the soundness and validity of the experimental design and analysis. Overall the experimental setup is well-structured.
However, the paper does not compare PF3plat to a simple pipeline using Mast3R for pose estimation + MVSplat for rendering. Since Mast3R outperforms PF3plat in pose estimation, it is unclear whether a Mast3R + MVSplat baseline would yield better results than PF3plat.
Without this comparison, it’s hard to judge whether PF3plat’s pose-free approach is necessary or if it just introduces more error.

**Methods And Evaluation Criteria:**

The benchmarks make sense for the problem. The paper tackles a well-motivated problem: most existing 3DGS methods rely on accurate camera poses, which are difficult to obtain in casual image capture scenarios. PF3plat removes this dependency, making it more practical for real-world applications.

However, the method still requires ground-truth (GT) camera intrinsics, which may not always be available in real-world scenarios. This contradicts the claim of being fully pose-free since intrinsic parameters are a part of the camera model.

**Other Comments Or Suggestions:**

1. Overall the paper is well-written and easy-to-follow. There are some typos, e.g. "fast speed".
2. Equation (2) does not clearly define E_pos. Equation (3) should use consistent notation for confidence scores.
3. The caption in Fig.1 should briefly explain the input/output of each module.

**Other Strengths And Weaknesses:**

Strengths:
1) The coarse alignment module creatively uses monocular depth estimation + feature matching to initialize 3D Gaussian positions. The method is fully feed-forward, making it more scalable and applicable to real-world settings.
2) The paper introduces confidence-aware refinement, ensuring that unreliable depth/pose estimates are given lower weight when estimating 3D Gaussians. This helps stabilize training and improves robustness.

Weakness:
a) ACID dataset results indicate weaknesses in large-scale outdoor scenes.
b) PF3plat claims robustness in low-texture regions, but this is unclear.

**Questions For Authors:**

1. The explanation of how the refined poses are computed is unclear (Sec. 3.2.3).
2. In Sec 3.2.4, about Cost Volume Construction and Aggregation, how does this differ from prior multi-view stereo (MVS) cost volumes?

**Relation To Broader Scientific Literature:**

PF3plat extends prior works (e.g. PixelSplat, MVSplat) by:
Removing the need for ground-truth camera poses, which prior 3DGS methods required.
Introducing coarse-to-fine pose estimation using monocular depth and correspondence networks, making pose-free 3DGS feasible.
Using confidence-aware refinement to stabilize Gaussian placement, which was a known issue in pixel-aligned 3DGS.

**Theoretical Claims:**

The paper does not appear to contain formal theoretical proofs—it is primarily an experimental and algorithmic contribution focused on pose-free novel view synthesis using 3DGS. Since there are no complex theoretical derivations or proofs, there are no major mathematical errors to verify.

---

> ### Author Rebuttal · Authors · 2025-04-01
>
> > The paper states that correspondence models like LightGlue struggle in low-texture regions. It is unclear how PF3plat can improve in these areas.
>
> We enhance performance in low-texture regions through two complementary approaches:
> As explained in line 233, our proposed 2D-3D consistency loss encourages corresponding points to lie on the same surface, drawing from principles of multi-view geometry. The effectiveness is shown in ***Table 4 (I-III)***.
>
> Second, the flexibility of the design choice at coarse alignment stage allows us to use other methods beyond LightGlue. For example, replacing LightGlue with networks like RoMA has led to noticeable improvements, as demonstrated in ***Table 6***.
>
> > Lack of qualitative results for pose estimation.
>
> Please refer to ***Figure 7***, which provides a visual comparison for pose estimation. Additionally, we have included the baseline pose estimates in this figure to clearly demonstrate the effectiveness of our fine alignment module. Please refer to the following anonymous link: https://anonymous.4open.science/r/7048/7048_camvis.pdf
>
> > However, the method still requires GT camera intrinsics, which may not always be available in real-world scenarios.
>
> Camera intrinsics are commonly required in previous pose-free methods, such as DBARF, CoPoNeRF and NoPoSplat, as  discussed in Sec. A.6. While this is a common limitation, for this rebuttal, we have conducted an additional experiment using camera intrinsics from UniDepth prediction:
>
> | Method                 | PSNR    | SSIM   | LPIPS |
> |------------------------|---------|--------|-------|
> | CoPoNeRF               | 19.536  |  0.638 | 0.398 |
> | Ours w/o GT intrinsics | 22.688  | 0.733  | 0.201 |
> | Ours                   | 23.589  | 0.782  | 0.181 |
>
> From the results, we highlight that even without GT intrinsics, ours outperforms CoPoNeRF and performs on par with the model with intrinsics. This suggests a potentially feasible approach to alleviate the limitation.
>
> > Mast3R for pose estimation + MVSplat for rendering.
>
> We opted not to use MVSplat with MASt3R because ***MASt3R's runtime of around 10 seconds conflicts with our goal of achieving a fast, feed-forward process***. Despite our superior performance in in RealEstate10K, thanks to the relatively larger-scale outdoor training of MASt3R, it is true that MASt3R performs slightly better than ours in pose estimation in ACID. Nevertheless, this is compensated when compared to MAST3R* in Tab. 2. We also highlight that with more advanced correspondence network, RoMa, ours further broadens the gap, as shown in Tab. 6.
>
> Nonetheless, for this rebuttal, we provide an additional comparison below:
> |                           |         |       |       |       |
> |---------------------------|---------|-------|-------|-------|
> | Method                    | PSNR    | SSIM  | LPIPS | Time  |
> | (0) Coarse Pose + MVSplat | 20.140  | 0.694 | 0.281 | 0.264 |
> | Mast3R + MVSplat          | 21.712  | 0.721 | 0.254 | 11    |
> | Mast3R* + MVSplat         | 21.167  | 0.702 | 0.268 | 0.642 |
> | Ours                      | 23.589  | 0.782 | 0.181 | 0.390 |
>
> From the results, we find that ours outperforms the baselines, thanks to our fine alignment modules and confidence estimation that contribute to further improvement.
>
> > ACID dataset results indicate weaknesses in large-scale outdoor scenes
>
> We acknowledge that our performance in large-scale outdoor scenes (dynamic coastline environments), is not optimal, ***as discussed in Sec. A.6***. We attribute these challenges primarily to UniDepth's training dataset, which, unlike DUSt3R or MASt3R, was built on a smaller and less diverse collection of outdoor scenes. A straightforward approach would be to leverage depth models that are trained on large-scale outdoor scenes or advanced correspondence models, to yield more accurate initial 3D Gaussian locations.
>
> > The explanation of how the refined poses are computed is unclear
>
> In Section 3.2.3, the refined pose is computed using three distinct inputs: Plücker Coordinates,  Feature Maps and Pose Token. Each of these inputs is processed through a series of attention layers, which help propagate information about previous camera estimates, multi-view geometry, and the current camera state. After fusing this information, the pose token is passed through a simple MLP that predicts residual rotation and translation parameters. These residuals are then added to the coarse camera parameters to yield the final refined pose.
>
> > Cost Volume Construction and Aggregation, how does this differ from prior multi-view stereo cost volumes?
>
> Our approach differs from traditional MVS cost volumes by introducing a guidance cost volume. Since the cost volume built from our estimated camera poses is inherently noisy, we supplement it with a guidance cost volume derived from a monocular depth estimate. By aggregating these two volumes, we construct a fina cost volume better tailored for a pose-free setting.

---

> > ### Comment · Reviewer_TAPU · 2025-04-07
> >
> > Thanks to the authors for the detailed explanation. The comments partially solved my problems. However, my main concern about the overall reconstruction quality still exists. From the visualizations in main paper and supplementary materials, the reconstructed scenes contain limited view change and blurry/floating regions. I also suggest use a higher resolution to improve visual quality. So, I maintain my original score.

---

> > > ### Author Response · Authors · 2025-04-09
> > >
> > > We appreciate the reviewer's feedback and apologize for not sufficiently addressing the concerns regarding the limited viewpoint variations and visual artifacts such as blurry or floating regions in our original submission. To clearly demonstrate the capability of our method under challenging scenarios involving large viewpoint changes, we provide additional visualizations through the following link (Please download the file, since the anonymous github seems to have a bug, not showing the captions properly) :
> > >
> > > https://anonymous.4open.science/r/7048/7048_qual.pdf
> > >
> > > These new visualizations explicitly include samples exhibiting significant viewpoint shifts, where our method ***consistently outperforms existing state-of-the-art methods*** such as DBARF, FlowCAM, and CoPoNeRF. Notably, our method produces higher-quality rendered images even under conditions of minimal overlap between context images. Given this clear demonstration of superiority, we respectfully suggest it would be unreasonable to maintain a rejection rating solely based on the originally perceived quality issue.
> > >
> > > We emphasize that although the absolute quality of our renderings may not match that of pose-supervised or pose-required approaches and does not render higher-resolution images (requires additional fine-tuning), ***this comparison should be contextualized within the scope of our task***. Our contribution specifically addresses the highly challenging scenario of pose-free, feed-forward novel view synthesis, where ground-truth poses are leveraged neither during training nor inference. Therefore, expecting similar image quality to supervised or pose-based methods would not be entirely fair or appropriate.
> > >
> > > We hope our response adequately addresses reviewer's concern.

---

### Decision · Program_Chairs · 2025-05-01

**Decision:**

Accept (poster)

**Comment:**

The paper initially received mixed scores (2,2,3,3,3) and was on the fence. The authors submitted a rebuttal which did not help clarify the main concerns of the reviewers leaning towards rejection, in particular around the novelty of the work and the limited quality of the results provided. At the same time, one other reviewer confirmed the support for the acceptance based on the performance demonstrated by the method. Two other reviewers did not provide further feedback, despite the reminders from the AC. Taken all comments and material into account, the current AC recommendation is to weak accept this work. This decision has been also reviewed and agreed upon by the SAC.